# Rates of Estimation of Optimal Transport Maps using Plug-in Estimators via Barycentric Projections

**Nabarun Deb**
Columbia University

**Promit Ghosal**
MIT

**Bodhisattva Sen**
Columbia University

## Abstract

Optimal transport maps between two probability distributions $\mu$ and $\nu$ on $\mathbb{R}^d$ have found extensive applications in both machine learning and statistics. In practice, these maps need to be estimated from data sampled according to $\mu$ and $\nu$. Plug-in estimators are perhaps most popular in estimating transport maps in the field of computational optimal transport. In this paper, we provide a comprehensive analysis of the rates of convergences for general plug-in estimators defined via barycentric projections. Our main contribution is a new stability estimate for barycentric projections which proceeds under minimal smoothness assumptions and can be used to analyze general plug-in estimators. We illustrate the usefulness of this stability estimate by first providing rates of convergence for the natural discrete-discrete and semi-discrete estimators of optimal transport maps. We then use the same stability estimate to show that, under additional smoothness assumptions of Sobolev type or Besov type, kernel smoothed or wavelet based plug-in estimators respectively speed up the rates of convergence and significantly mitigate the curse of dimensionality suffered by the natural discrete-discrete/semi-discrete estimators. As a by-product of our analysis, we also obtain faster rates of convergence for plug-in estimators of $W_2(\mu, \nu)$, the Wasserstein distance between $\mu$ and $\nu$, under the aforementioned smoothness assumptions, thereby complementing recent results in Chizat et al. (2020). Finally, we illustrate the applicability of our results in obtaining rates of convergence for Wasserstein barycenter between two probability distributions and obtaining asymptotic detection thresholds for some recent optimal-transport based tests of independence.

## 1   Introduction

Given two random variables $X \sim \mu$ and $Y \sim \nu$, where $\mu, \nu$ are probability measures on $\mathbb{R}^d$, $d \geq 1$, the problem of finding a "nice" map $T_0(\cdot)$ such that $T_0(X) \sim \nu$ has numerous applications in machine learning such as domain adaptation and data integration [34, 35, 38, 48, 61, 112], dimension reduction [12, 66, 90], generative models [60, 81, 88, 110], to name a few. Of particular interest is the case when $T_0(\cdot)$ is obtained by minimizing a cost function, a line of work initiated by Gaspard Monge [97] in 1781 (see (1.1) below), in which case $T_0(\cdot)$ is termed an *optimal transport* (OT) map and has applications in shape matching/transfer problems [29, 47, 107, 121], Bayesian statistics [46, 75, 80, 108], econometrics [15, 28, 45, 50, 54], nonparametric statistical inference [39–41, 113, 114]; also see [111, 128, 129] for book-length treatments on the subject. In this paper, we will focus on the OT map obtained using the standard squared Euclidean cost function, i.e.,

$$T_0 := \underset{T:T\#\mu=\nu}{\operatorname{argmin}} \mathbb{E}\|X - T(X)\|^2, \tag{1.1}$$

where $T\#\mu = \nu$ means $T(X) \sim \nu$ for $X \sim \mu$. The estimation of $T_0$ has attracted a lot of interest in recent years due to its myriad applications (as stated above) and interesting geometrical properties (see [19, 56, 91] and Definition 1.1 below). In practice, the main hurdle in constructing estimators for $T_0$ is that the explicit forms of the measures $\mu, \nu$ are unknown; instead only random samples

$$X_1, \ldots, X_m \sim \mu \qquad \text{and} \qquad Y_1, \ldots, Y_n \sim \nu$$

35th Conference on Neural Information Processing Systems (NeurIPS 2021).

are available. A natural strategy in this scenario is to estimate $T_0$ using $\widetilde{T}_{m,n}$, where $\widetilde{T}_{m,n}$ is computed as in (1.1) with $\mu$ and $\nu$ replaced by $\widetilde{\mu}_m$ and $\widetilde{\nu}_n$ which are empirical approximations of $\mu$ and $\nu$ based on $X_1, \ldots, X_m$ and $Y_1, \ldots, Y_n$ respectively (see Definition 1.2). Such estimators are often called *plug-in estimators* and have been used extensively; see [7, 30, 67, 93, 94, 102, 116].

The main goal of this paper is to study the *rates of convergence* of general plug-in estimators of $T_0$ under a unified framework. We show that when $\widetilde{\mu}_m$ and $\widetilde{\nu}_n$ are chosen as $\widehat{\mu}_m$ and $\widehat{\nu}_n$ respectively, where $\widehat{\mu}_m$ and $\widehat{\nu}_n$ are the standard empirical distributions supported on $m$ and $n$ atoms, i.e.,

$$\widehat{\mu}_m := \frac{1}{m} \sum_{i=1}^{m} \delta_{X_i} \qquad \text{and} \qquad \widehat{\nu}_n := \frac{1}{n} \sum_{j=1}^{n} \delta_{Y_j}, \tag{1.2}$$

$\widetilde{T}_{m,n}$ (appropriately defined using Definition 1.2) converges at a rate of $m^{-2/d} + n^{-2/d}$ for $d \geq 4$ in the sense of (1.8). This rate happens to be minimax optimal under minimal smoothness assumptions (see [72, Theorem 6]) but suffers from the *curse of dimensionality*. We next show that, if $\mu$ and $\nu$ are known to admit sufficiently smooth densities, it is possible to apply kernel or wavelet based smoothing techniques on $\widehat{\mu}_m$ and $\widehat{\nu}_n$ to obtain plug-in estimators that mitigate the aforementioned curse of dimensionality.

Our next contribution pertains to the estimation of $W_2^2(\mu, \nu)$ (the squared Wasserstein distance), see (1.3) below, a quantity of independent interest in statistics and machine learning with applications in structured prediction [51, 89], image analysis [18, 59], nonparametric testing [16, 106], generative modeling [10, 96], etc. In this paper, we also obtain rates of convergence for plug-in estimators $W_2^2(\widetilde{\mu}_m, \widetilde{\nu}_n)$ of $W_2^2(\mu, \nu)$. We show that kernel smoothing $\widehat{\mu}_m$ and $\widehat{\nu}_n$ can be used to obtain plug-in estimators of $W_2^2(\mu, \nu)$ that mitigate the curse of dimensionality as opposed to a direct plug-in approach using $\widehat{\mu}_m$ and $\widehat{\nu}_n$ (as used in [30, Theorem 2]). This provides an answer to the open question of estimating $W_2^2(\mu, \nu)$ when $\mu, \nu$ admit smooth densities laid out in [30].

## 1.1 Background on optimal transport

In this section, we present some basic concepts and results associated with the OT problem that will play a crucial role in the sequel. Let $\mathcal{P}_{\mathrm{ac}}(\mathbb{R}^d)$ denote the set of all Lebesgue absolutely continuous probability measures on $\mathbb{R}^d$ and $\mathcal{P}_2(\mathbb{R}^d)$ be the set of probability measures with finite second moments. Then the 2-*Wasserstein* distance (squared) between $\mu, \nu \in \mathcal{P}_2(\mathbb{R}^d)$ is defined as:

$$W_2^2(\mu, \nu) := \min_{\pi \in \Pi(\mu, \nu)} \int \|x - y\|^2 \, d\pi(x, y), \tag{1.3}$$

where $\Pi(\mu, \nu)$ is the set of probability measures on $\mathbb{R}^d \times \mathbb{R}^d$ with marginals $\mu$ and $\nu$. The optimization problem in (1.3) is often called the *Kantorovich relaxation* (see [76, 77]) of the optimization problem in (1.1). The existence of a minimizer in (1.3) follows from [129, Theorem 4.1].

**Proposition 1.1** (Brenier-McCann polar factorization theorem, see [91, 128]). *Suppose $\mu \in \mathcal{P}_{\mathrm{ac}}(\mathbb{R}^d)$. Then there exists a $\mu$-a.e. (almost everywhere) unique function $T_0(\cdot) : \mathbb{R}^d \to \mathbb{R}^d$, which is the gradient of a real-valued $d$-variate convex function, say $\varphi_0(\cdot) : \mathbb{R}^d \to \mathbb{R}$, such that $T_0 \# \mu = \nu$. Further, the distribution defined as $\pi(A \times B) = \mu(A \cap (T_0)^{-1}(B))$ for all Borel sets $A, B \subseteq \mathbb{R}^d$ is the unique minimizer in (1.3) provided $\mu, \nu \in \mathcal{P}_2(\mathbb{R}^d)$.*

**Definition 1.1** (OT map and potential function). *The function $T_0 : \mathbb{R}^d \to \mathbb{R}^d$ in Proposition 1.1 which satisfies $T_0 \# \mu = \nu$ will be called the OT map from $\mu$ to $\nu$. A convex function $\varphi_0(\cdot)$ in Proposition 1.1 satisfying $\nabla \varphi_0 = T_0$ will be termed an OT potential.*

The next and final important ingredient is the alternate dual representation of (1.3) which gives:

$$\frac{1}{2} W_2^2(\mu, \nu) = \frac{1}{2} \int \|x\|^2 \, d\mu(x) + \frac{1}{2} \int \|y\|^2 \, d\nu(y) - \min_{f \in \mathcal{F}} \mathcal{S}_{\mu, \nu}(f), \qquad \text{where} \tag{1.4}$$

$$\mathcal{S}_{\mu, \nu}(f) = \int f \, d\mu + \int f^* \, d\nu. \tag{1.5}$$

Here $\mathcal{F}$ denotes the space of convex functions on $\mathbb{R}^d$ which are also elements of $L^1(\mu)$ and $f^*(\cdot)$ is the standard Legendre-Fenchel dual defined as:

$$f^*(x) := \sup_{y \in \mathbb{R}^d} [y^\top x - f(y)], \qquad \text{for } x \in \mathrm{dom}(f). \tag{1.6}$$

## 1.2 Estimating OT map via barycentric projection

Recall the setting from the Introduction. Let $\widetilde{\mu}_m, \widetilde{\nu}_n \in \mathcal{P}_2(\mathbb{R}^d)$. Here $\widetilde{\mu}_m, \widetilde{\nu}_n$ need not be absolutely continuous and can be very general. Intuitively, $\widetilde{\mu}_m$ and $\widetilde{\nu}_n$ can be viewed as some empirical approximation of $\mu$ and $\nu$ respectively.

**Example 1.2** (Simple choices of $\widetilde{\mu}_m$ and $\widetilde{\nu}_n$). *Let $X_1, \ldots, X_m \overset{i.i.d.}{\sim} \mu$ and $Y_1, \ldots, Y_n \overset{i.i.d.}{\sim} \nu$; in which case a natural choice would be to set $\widetilde{\mu}_m = \widehat{\mu}_m$ and $\widetilde{\nu}_n = \widehat{\nu}_n$ where $\widehat{\mu}_m$ and $\widehat{\nu}_n$ are the empirical distributions of $X_1, \ldots, X_m$ and $Y_1, \ldots, Y_n$ respectively, as defined in* (1.2). *This is the standard choice adopted in the discrete-discrete Kantorovich relaxation; see [104, Section 2.3]. Another popular choice is $\widetilde{\mu}_m = \widehat{\mu}_m$, $\widetilde{\nu}_n = \nu$ or $\widetilde{\mu}_m = \mu$, $\widetilde{\nu}_n = \widehat{\nu}_n$. This is the semi-discrete Kantorovich problem and is popular when one of the measures is fully specified; see [26, 55].* ∎

A natural way to estimate $T_0(\cdot)$, as defined in (1.1), would be to approximate it using the OT map from $\widetilde{\mu}_m$ to $\widetilde{\nu}_n$. However as $\widetilde{\mu}_m$ and $\widetilde{\nu}_n$ may not be elements of $\mathcal{P}_{\mathrm{ac}}(\mathbb{R}^d)$, Proposition 1.1 does not apply and an OT map *may not exist* from $\widetilde{\mu}_m$ to $\widetilde{\nu}_n$. Such is the case in Example 1.2 in the *discrete-discrete* case when $m \neq n$. To circumvent this issue, we leverage the notion of barycentric projections (see [3, Definition 5.4.2]) defined below:

**Definition 1.2** (Barycentric projection). *Define the set*

$$\widetilde{\Gamma}_{\min} := \underset{\pi \in \Pi(\widetilde{\mu}_m, \widetilde{\nu}_n)}{\operatorname{argmin}} \int \|x - y\|^2 \, d\pi(x, y).$$

*The optimization problem above is the plug-in analog of the optimization problem on the right hand side of* (1.3). *Given any $\gamma \in \widetilde{\Gamma}_{\min}$, define the barycentric projection of $\gamma$ as the conditional mean of $y$ given $x$ under $\gamma$, i.e.,*

$$\widetilde{T}_{m,n}(x) \equiv \widetilde{T}_{m,n}^{\gamma}(x) := \frac{\int_y y \, d\gamma(x, y)}{\int_y d\gamma(x, y)}, \qquad \text{for } x \in supp\left(\widetilde{\mu}_m\right). \tag{1.7}$$

In general, $\widetilde{\Gamma}_{\min}$ need not be a singleton which is why we index the barycentric projection $\widetilde{T}_{m,n}^{\gamma}(\cdot)$ by $\gamma \in \widetilde{\Gamma}_{\min}$. Note that $\widetilde{T}_{m,n}^{\gamma}(\cdot)$ need not be a transport map; however, if an OT map exists then it must be equal to $\widetilde{T}_{m,n}^{\gamma}(\cdot)$ ($\widetilde{\mu}_m$-a.e.). Our goal is to obtain stochastic upper bounds for

$$\sup_{\gamma \in \widetilde{\Gamma}_{\min}} \int \left\|\widetilde{T}_{m,n}^{\gamma}(x) - T_0(x)\right\|^2 d\widetilde{\mu}_m(x). \tag{1.8}$$

In addition, our proof techniques also yield rates of convergence for

$$\left| W_2^2(\widetilde{\mu}_m, \widetilde{\nu}_n) - W_2^2(\mu, \nu) \right|. \tag{1.9}$$

In this paper, we will focus on $d \geq 2$. Due to the canonical ordering of $\mathbb{R}$, the case $d = 1$ can be handled easily using the classical Hungarian embedding theorem [82].

## 1.3 Contributions

1. We provide a new and flexible stability estimate Theorem 2.1 which yields a unified approach to obtaining *rates* of convergence for general plug-in estimators of the OT map $T_0(\cdot)$. Unlike existing stability estimates, Theorem 2.1 holds for the barycentric projection (which is the same as the OT map when it exists) and does not require any smoothness assumptions on $\widetilde{\mu}_m, \widetilde{\nu}_n$ or $\widetilde{T}_{m,n}^{\gamma}(\cdot)$; also see Remark 2.1 for a comparison with the existing literature.

2. In Sections 2.1 and 2.2, we use Theorem 2.1 to bound (1.8) and (1.9):

   - In Section 2.1, we show that in both the discrete-discrete and semi-discrete Kantorovich relaxation problems (see Example 1.2), the rate of convergence of (1.8) is $m^{-2/d} + n^{-2/d}$ for $d \geq 4$ when $T_0$ is assumed to be Lipschitz (see Theorem 2.2), which is the minimax rate (see [72, Theorem 6]). To the best of our knowledge, rates of convergence for these natural estimators weren't previously established in the literature.

- In Section 2.2 and Appendix A, we show that the curse of dimensionality in the above rates can be mitigated provided $\mu$ and $\nu$ admit (uniform) Sobolev smooth densities (see Section 2.2) or Besov smooth densities (see Appendix A). In Section 2.2, our plug-in estimator is obtained by choosing $\widetilde{\mu}_m$ (and $\widetilde{\nu}_n$) as the convolution of $\widehat{\mu}_m$ (and $\widehat{\nu}_n$) and a smooth kernel with an appropriate bandwidth. Under this choice, the rate of convergence in (1.8) is $m^{-\left(\frac{s+2}{d}\wedge\frac{1}{2}\right)} + n^{-\left(\frac{s+2}{d}\wedge\frac{1}{2}\right)}$, where $s$ denotes the degree of Sobolev smoothness (see Theorem 2.5). Clearly, if $2(s+2) \geq d$, the rate of convergence becomes dimension-free and mitigates the curse of dimensionality. We also show the same rates of convergence mentioned above hold for (1.9) (see e.g., Proposition 2.6) which makes a strong case in favor of incorporating smoothness in the construction of plug-in estimators as was conjectured in [30]. In Appendix A, our plug-in estimator is obtained using natural wavelet based density estimators. The rate of convergence in (1.8) turns out to be $n^{-\frac{1+s}{d+2s}}$ where $s$ denotes the degree of Besov smoothness (see Theorem A.1). Note that by choosing $s$ large enough, the exponent in the rate can be made arbitrarily close to $1/2$, thereby reducing the curse of dimensionality.

3. In Section 2.3, we use a discretization technique from [131] to construct discrete approximations to the smoothed $\widetilde{\mu}_m$ and $\widetilde{\nu}_n$ from the previous paragraph that in turn yield computable plug-in estimators for $T_0$ (provided one can sample from $\widetilde{\mu}_m$ and $\widetilde{\nu}_n$) that also achieve the same statistical guarantees as the smoothed plug-in estimator from Section 2.2 (see Theorem 2.7). However the number of atoms required in the discretizations and correspondingly the computational complexity increases with the degree of smoothness; this highlights a statistical and computational trade-off.

4. We provide implications of our results in popular applications of OT such as estimating the barycenter of two multivariate probability distributions (see Theorem B.1 in Appendix B.1) and in nonparametric independence testing (see Theorem B.3 in Appendix B.2).

## 1.4 Related work

Many recent works have focused on obtaining consistent estimators of $T_0$ using the plug-in principle, see [26, 55] (in the semi-discrete problem) and [41, 68, 132] (in the discrete-discrete problem). In [55], the authors studied the rate of convergence of the semi-discrete optimal transport map from $\nu$ (absolutely continuous) to $\widehat{\mu}_m$. This paper complements the aforementioned papers by studying the rates of convergence for general plug-in estimators in a unified fashion. In two other papers [9, Theorem 1.1] and [87, Section 4], the authors use a "Voronoi tessellation" approach to estimate $T_0$, however the rates obtained in this paper, even in the absence of smoothness, are strictly better than those in [9, 87]. Perhaps the most closely related paper to ours would be [67]. In [67], the author uses variational techniques to arrive at stability estimates while we exploit the Lipschitz nature of the OT map (see Definition 1.1). Further the rates in this paper have exponents $\frac{s+2}{d} \wedge \frac{1}{2}$ which are *strictly better* than the exponents $\frac{s+2}{2(s+2)+d}$ obtained in [67, Proposition 1] under the same smoothness assumptions (Sobolev type of order $s$, see Definition 2.4). In another line of work [72], the authors use theoretical wavelet based estimators (not of the plug-in type) of $T_0$ to obtain nearly minimax optimal rates of convergence. However these estimators, by themselves, are not transport maps between two probability measures, which makes them harder to interpret. In contrast, our focus is on obtaining rates of convergence for plug-in estimators, which are transport maps between natural approximations of $\mu$ and $\nu$. Such plug-in type strategies are a lot more popular in computational OT [7, 30, 67, 93, 94, 102, 116].

In terms of obtaining rates of convergence for (1.9), some attempts include [109, 116] where parametric rates are obtained when $\mu, \nu$ are known to be finitely supported or are both Gaussian. In a related problem, bounds for $W_2^2(\widehat{\mu}_m, \mu)$ were obtained in [6, 42, 49, 100, 123, 131]. Using these bounds, for $m = n$, it is easy to get a $n^{-1/d}$ rate of convergence for (1.9). This rate was recently improved to $n^{-2/d}$ in [30] under no smoothness assumptions. Our rates coincide with the $n^{-2/d}$ rate from [30] under no smoothness assumptions. But further, we show in this paper that the curse of dimensionality in the above rate can be mitigated by incorporating smoothness into the plug-in procedure.

## 2 Main results

Recall $\varphi_0(\cdot)$ from Definition 1.1. The following is our main result.

**Theorem 2.1** (Stability estimate). *Suppose that $\mu, \nu \in \mathcal{P}_{ac}(\mathbb{R}^d) \cap \mathcal{P}_2(\mathbb{R}^d)$ and $\widetilde{\mu}_m, \widetilde{\nu}_n \in \mathcal{P}_2(\mathbb{R}^d)$. Assume that $T_0(\cdot)$ (as defined in (1.1)) is L-Lipschitz $(L > 0)$. Then,*

$$\sup_{\gamma \in \widetilde{\Gamma}_{\min}} \int \|\widetilde{T}_{m,n}^{\gamma}(x) - T_0(x)\|^2 \, d\widetilde{\mu}_m(x) \leq L \max \left\{ \left| \int \Psi_{\widetilde{\mu}_m, \widetilde{\nu}_n}^* \, d(\widetilde{\nu}_n - \nu_m^{\dagger}) \right|, \left| \int \Psi_{\widetilde{\mu}_m, \nu_m^{\dagger}}^* \, d(\widetilde{\nu}_n - \nu_m^{\dagger}) \right| \right\}$$

$$+ 2L \int \varphi_0^*(y) \, d(\widetilde{\nu}_n - \nu_m^{\dagger})(y), \tag{2.1}$$

*where $\nu_m^{\dagger} := T_0 \# \widetilde{\mu}_m$, $\varphi_0^*(\cdot)$ is defined as in (1.6), and with $\mathcal{S}_{\cdot,\cdot}(\cdot)$ defined as in (1.5), $\Psi_{\widetilde{\mu}_m, \widetilde{\nu}_n}^*(\cdot) := \operatorname{argmin}_{f \in \mathcal{F}} \mathcal{S}_{\widetilde{\mu}_m, \widetilde{\nu}_n}(f)$, $\Psi_{\widetilde{\mu}_m, \nu_m^{\dagger}}^*(\cdot) := \operatorname{argmin}_{f \in \mathcal{F}} \mathcal{S}_{\widetilde{\mu}_m, \nu_m^{\dagger}}(f)$, and $\mathcal{D}$ denotes the space of real-valued convex functions on $\mathbb{R}^d$.*

The proof of Theorem 2.1 (see Appendix C.1) starts along the same lines as the proof of the curvature estimate in [56, Proposition 3.3]. This is followed by some careful manipulations of $W_2^2(\cdot, \cdot)$ (as in (1.3)) and an application of the conditional version of Jensen's inequality, see (C.3). The final step of the proof uses the dual representation in (1.4) with techniques similar to some intermediate steps in the proof of [92, Proposition 2] and [30, Lemma 3].

**Remark 2.1** (Comparison with other stability estimates). *Theorem 2.1 provides some important advantages to existing stability estimates in the literature. One of the earliest results in this direction can be found in [56, Proposition 3.3] but their bound involves a push-forward constraint which makes it hard to use for rate of convergence analysis. A bound similar to Theorem 2.1 is presented in [55, Lemma 5.1] but there the authors assume the existence of an OT map from $\widetilde{\mu}_m$ to $\widetilde{\nu}_n$. Therefore, it does not apply to the discrete-discrete problem where $\widetilde{\mu}_m = \widehat{\mu}_m$ and $\widetilde{\nu}_n = \widehat{\nu}_n$ with $m \neq n$. Overcoming all these limitations is an important contribution of Theorem 2.1 and allows us to deal with popular plug-in estimators all in one go. The stability estimate in [72, Proposition 10] on the other hand requires $\widetilde{\mu}_m, \widetilde{\nu}_n$ to be sufficiently smooth and hence it does not hold for discrete-discrete or semi-discrete plug-in estimators (see Example 1.2). Further their result requires all the measures involved to be compactly supported unlike the much milder requirements of Theorem 2.1. However, a shortcoming of Theorem 2.1 is that it is hard to obtain rates faster than $n^{-1/2}$ using it directly, whereas [72] can obtain rates arbitrarily close to $n^{-1}$. This is a price we pay for analyzing natural and popular plug-in estimators as opposed to the (more intractable) wavelet based estimators in [72].*

**Remark 2.2** (*How to use Theorem 2.1 to obtain rates of convergence?*). *Note that the second term on the right hand side of* (2.1), *under appropriate moment assumptions, is $O_p(m^{-1/2} + n^{-1/2})$ (free of dimension) by a direct application of Markov's inequality. We therefore focus on the first term. By* (1.5), *$\Psi_{\widetilde{\mu}_m, \widetilde{\nu}_n}^*(\cdot)$, $\Psi_{\widetilde{\mu}_m, \nu_m^{\dagger}}^*(\cdot) \in \mathcal{F}$. Further, by Caffarelli's regularity theory [20–22], depending on the "smoothness" of $\widetilde{\mu}_m, \widetilde{\nu}_n$, it can be shown that there exists a further class of functions $\mathcal{F}_s$ (see Remarks 2.3 and 2.6) such that $\Psi_{\widetilde{\mu}_m, \widetilde{\nu}_n}^*(\cdot)$, $\Psi_{\widetilde{\mu}_m, \nu_m^{\dagger}}^*(\cdot) \in \mathcal{F} \cap \mathcal{F}_s$. Thus, we can bound the first term on the right hand side of* (2.1) *as:*

$$\max \left\{ \left| \int \Psi_{\widetilde{\mu}_m, \widetilde{\nu}_n}^* \, d(\widetilde{\nu}_n - \nu_m^{\dagger}) \right|, \left| \int \Psi_{\widetilde{\mu}_m, \nu_m^{\dagger}}^* \, d(\widetilde{\nu}_n - \nu_m^{\dagger}) \right| \right\} \leq \sup_{f \in \mathcal{F} \cap \mathcal{F}_s} \left| \int f \, d(\widetilde{\nu}_n - \nu_m^{\dagger}) \right|. \tag{2.2}$$

*The right hand side of* (2.2) *can now be bounded using the corresponding Dudley's entropy integral bounds using empirical process techniques; see [126, Lemmas 19.35-19.37].*

To conclude, the two main steps in our strategy are identifying the family of functions $\mathcal{F}_s$ and computing Dudley's entropy integral. Further, the more the smoothness of $\widetilde{\mu}_m, \widetilde{\nu}_n$, the smaller is the class of functions $\mathcal{F}_s$ and smaller the supremum on the right hand side of (2.2). This shows why better rates can be expected under smoothness assumptions.

## 2.1 Natural non-smooth plug-in estimators

In this case, we discuss the rates of convergence for the discrete-discrete problem and the semi-discrete problem, where *no smoothness* is available on $\widetilde{\mu}_m$ and $\widetilde{\nu}_n$.

**Theorem 2.2.** *Suppose that $T_0(\cdot)$ is L-Lipschitz, $\nu$ is compactly supported and $\mathbb{E} \exp(t\|X_1\|^{\alpha}) < \infty$ for some $t > 0$, $\alpha > 0$.*

*(Discrete-discrete):* Set $\widetilde{\mu}_m = \widehat{\mu}_m$ and $\widetilde{\nu}_n = \widehat{\nu}_n$. Then the following holds:

$$\sup_{\gamma \in \widetilde{\Gamma}_{\min}} \int \|\widetilde{T}^\gamma_{m,n}(x) - T_0(x)\|^2 \, d\widetilde{\mu}_m(x) = O_p\left(r_d^{(m,n)} \times (\log(1 + \max\{m,n\}))^{t_{d,\alpha}}\right), \quad (2.3)$$

$$\text{where} \quad r_d^{(m,n)} := \begin{cases} m^{-1/2} + n^{-1/2} & \text{for } d = 2,3, \\ m^{-1/2}\log(1+m) + n^{-1/2}\log(1+n) & \text{for } d = 4, \\ m^{-2/d} + n^{-2/d} & \text{for } d \geq 5, \end{cases} \quad (2.4)$$

*and*

$$t_{d,\alpha} := \begin{cases} (4\alpha)^{-1}(4 + ((2\alpha + 2d\alpha - d) \vee 0)) & \text{for } d < 4, \\ (\alpha^{-1} \vee 7/2) - 1 & \text{for } d = 4, \\ 2(1 + d^{-1}) & \text{for } d > 4. \end{cases}$$

*The same bound holds for $|W_2^2(\widetilde{\mu}_m, \widetilde{\nu}_n) - W_2^2(\mu, \nu)|$ without assuming $T_0(\cdot)$ is Lipschitz.*

*(Semi-discrete):* Set $\widetilde{\mu}_m = \mu$, $\widetilde{\nu}_n = \widehat{\nu}_n$ or $\widetilde{\mu}_m = \widehat{\mu}_m$, $\widetilde{\nu}_n = \nu$. Then the left hand side of (2.3) is $O_p(r_d^{(n,n)} \times (\log(1+n))^{t_{d,\alpha}})$ or $O_p(r_d^{(m,m)} \times (\log(1+m))^{t_{d,\alpha}})$ respectively.

A stronger result can be proved if both $\mu$ and $\nu$ are compactly supported.

**Corollary 2.3.** *Consider the setting from Theorem 2.2 and assume further that $\mu$ is compactly supported. Then, with $r_d^{(m,n)}$ defined as in (2.4), we have:*

$$\mathbb{E}\left[\sup_{\gamma \in \widetilde{\Gamma}_{\min}} \int \|\widetilde{T}^\gamma_{m,n}(x) - T_0(x)\|^2 \, d\widetilde{\mu}_m(x)\right] \leq C r_d^{(m,n)},$$

*for some constant $C > 0$, in both the discrete-discrete and semi-discrete settings from Theorem 2.2.*

A brief description of the proof technique of Theorem 2.2 using Theorem 2.1 is provided in Remark 2.3 below, and the actual proof is presented in Appendix C.1.

**Remark 2.3** (Proof technique). *The proof of Theorem 2.2 proceeds via the strategy outlined in Remark 2.2. We first show that $\mathcal{F}_s$ (see Remark 2.2) can be chosen as a certain sub-class of convex functions which are in $L^2(\nu)$. We then use Dudley's entropy integral type bounds which in turn requires the bracketing entropy [126, Page 270] of $\mathcal{F}_s$, recently proved in [83, Equation 26]. This strategy is slightly different from that used in the proof of [30, Theorem 2], where the authors assume that $\mu$ is compactly supported whereas we only assume the finiteness of $\mathbb{E} \exp(t\|X_1\|^\alpha)$ for some $t > 0$, $\alpha > 0$. The compactness assumption on $\mu$ allows one to further restrict $\mathcal{F}_s$ to the class of Lipschitz functions. This additional restriction does not seem to be immediate without the compactness assumption.*

As discussed in Section 1.3, the exponents obtained in Theorem 2.2 are minimax optimal, up to multiplicative logarithmic factors, under bare minimal smoothness assumptions (see [72, Theorem 6]). To the best of our knowledge, rates for the discrete-discrete case for $m \neq n$ and those for the semi-discrete case were not known previously in the literature. Our rates are also strictly better than those (for different estimators, based on space tessellations) obtained in [9, 87] and require less stringent assumptions than those in [30]. In the next section, we show how smoothness assumptions can be leveraged to mitigate the curse of dimensionality in Theorem 2.2.

## 2.2 Smooth kernel based plug-in estimator: mitigating the curse of dimensionality

In this section, we focus on kernel based density estimators for the probability densities associated with $\mu$ and $\nu$ (see [57, 58, 99, 103, 115]). We will show, using Theorem 2.1, that the corresponding estimators of $T_0(\cdot)$ achieve (near) dimension-free rates under sufficient smoothness assumptions.

We first introduce the Sobolev class of functions which we will exploit in this subsection to construct estimators that achieve rates of convergence which mitigate the curse of dimensionality under sufficient smoothness.

**Definition 2.4** (Uniform Sobolev class of functions). *Let $\Omega \subseteq \mathbb{R}^d$ and $f(\cdot)$ be uniformly continuous on $\Omega$ and admits uniformly continuous derivatives up to order $s$ on $\Omega$ for some $s \in \mathbb{N}$. For any $\mathfrak{m} := (m_1, \ldots, m_d) \in \mathbb{N}^d$, let*

$$\partial^{\mathfrak{m}} f := \frac{\partial}{\partial_{x_1}^{m_1}} \cdots \frac{\partial}{\partial_{x_d}^{m_d}} f, \quad |\mathfrak{m}| := \sum_{i=1}^{d} m_i.$$

*For any $k \le s$, we further define,*

$$\|f\|_{C^k(\Omega)} := \sum_{|\mathfrak{m}| \le k} \|\partial^{\mathfrak{m}} f\|_{L^\infty(\Omega)}.$$

*The space $C^s(\Omega)$ is defined as the set of functions $f(\cdot)$ for which $\|f\|_{C^k(\Omega)} < \infty$ for all $k \le s$.*

For this subsection, assume that $\mu$ and $\nu$ admit Sobolev smooth densities $f_\mu(\cdot)$ and $f_\nu(\cdot)$ in the uniform norm (see Definition 2.4 above). Given $\Omega \subseteq \mathbb{R}^d$ and $s \in \mathbb{N}$, let $C^s(\Omega)$ denote the set of Sobolev smooth functions on $\Omega$ of order $s$.

**Assumption (A1)** (Regularity of the densities). *Suppose that*

1. *$f_\mu$ and $f_\nu$ are supported on compact and convex subsets of $\mathbb{R}^d$, say $\mathcal{X}$ and $\mathcal{Y}$ respectively.*

2. *There exists $s, M > 0$ such that $f_\mu(\cdot) \in C^s(\mathcal{X}; M)$ and $f_\nu(\cdot) \in C^s(\mathcal{Y}; M)$ where $C^s(\mathcal{X}; M)$ is the space of real valued functions supported on $\mathcal{X}$ such that for all $f(\cdot) \in C^s(\mathcal{X}; M)$, we have $M^{-1} \le f(x) \le M$ for all $x \in \mathcal{X}$ and $\|f\|_{C^s(\mathcal{X})} \le M$. Here $\|\cdot\|_{C^s(\mathcal{X})}$ is the standard uniform Sobolev norm as defined in Definition 2.4. The space $C^s(\mathcal{Y}; M)$ is defined analogously.*

We now define our estimators for $f_\mu(\cdot)$ and $f_\nu(\cdot)$ using the standard kernel density estimation technique (see [125, Section 1.2]). Set

$$\widehat{f}_\mu(x) := \frac{1}{m h_m^d} \sum_{i=1}^{m} K_d \left( \frac{X_i - x}{h_m} \right), \tag{2.5}$$

for some bandwidth parameter $h_m > 0$ and $d$-variate kernel $K_d(\cdot)$. We assume that $K_d(\cdot)$ is the $d$-fold product of univariate kernels, i.e., there exists a kernel $K(\cdot)$ such that for $u = (u_1, \ldots, u_d) \in \mathbb{R}^d$, $K_d(u) = \prod_{i=1}^{d} K(u_i)$. We define $\widehat{f}_\nu(\cdot)$ similarly with the same univariate kernel and bandwidth.

**Assumption (A2)** (Regularity of the kernel). *Assume that $K(\cdot)$ is a symmetric, bounded, $s+1$ times differentiable kernel on $\mathbb{R}^d$ with all $s+1$ derivatives bounded and integrable. Further, suppose that $K(\cdot)$ is of order $2s + 2$, i.e.,*

$$\int u^j K(u) \, du = \mathbb{1}(j = 0), \quad for \ j = \{0, 1, 2, \ldots, 2s+1\}, \quad and \int |u|^{2s+2} |K(u)| \, du < \infty.$$

The above assumptions on $K(\cdot)$ are standard for estimating smooth densities and their derivatives of different orders in the kernel density estimation literature; see e.g. [4, 57, 58, 69, 125]. There are several natural ways to construct kernels satisfying Assumption (A2), see [125, Section 1.2.2]; an example is also provided in Example 2.4 below.

**Example 2.4** (Example of a kernel satisfying Assumption (A2)). *Let $\psi_m(\cdot)$ be the $m$-th Hermite polynomial on $\mathbb{R}$ (see [84]). Then the kernel function defined as*

$$K(u) := \sum_{m=0}^{2s+2} \psi_m(0) \psi_m(u) \exp(-u^2/2)$$

*satisfies Assumption (A2).* ∎

It is evident from Assumption (A2) that $K(\cdot)$ may take some negative values, in which case, $\widehat{f}_\mu(\cdot)$ (respectively $\widehat{f}_\nu(\cdot)$) may not be a probability density. Consequently the barycentric projection (see Definition 1.2) between $\widehat{f}_\mu(\cdot)$ and $\widehat{f}_\nu(\cdot)$ is not well-defined. We get around this by projecting

$\widehat{f}_\mu(\cdot)$ and $\widehat{f}_\nu(\cdot)$ on an appropriate space of "smooth" probability densities (see (2.6)), via an integral probability metric (see Definition 2.5 below; also see [98, 105, 117] for examples, computational procedures and applications of such metrics).

**Definition 2.5** (Integral probability metric). *Given a class $\mathcal{H}$ of bounded functions on $\mathbb{R}^d$ and two probability densities $g_1(\cdot)$ and $g_2(\cdot)$ on $\mathbb{R}^d$, the integral probability metric/distance between $g_1(\cdot)$ and $g_2(\cdot)$ with respect to $\mathcal{H}$ is defined as*

$$d_{\mathrm{IP}}(g_1, g_2; \mathcal{H}) := \sup_{\psi(\cdot) \in \mathcal{H}} \left| \int \psi(x)(g_1(x) - g_2(x))\, dx \right|.$$

*Sufficient conditions on $\mathcal{H}$ for $d_{\mathrm{IP}}(\cdot, \cdot; \mathcal{H})$ to be a metric on the space of probability measures (not on the space of probability densities as they can be altered on set of Lebesgue measure 0 without altering the underlying probability measures) on $\mathbb{R}^d$ have been discussed in [98]. Observe that the measure $d_{\mathrm{IP}}(g_1, g_2; \mathcal{H})$ is well defined even when $g_1(\cdot)$ and $g_2(\cdot)$ are not probability densities.*

*In Theorem 2.5 below, we use $\mathcal{H} = C^{s+2}(\mathcal{X}, M')$. Note that any function in $C^{s+2}(\mathcal{X}, M')$ can be extended to a function in $C^{s+2}(\mathbb{R}^d; M')$ (see [72, Theorem 23] and [124, Theorem 1.105]). The fact that this choice of $\mathcal{F}$ results in a metric follows from the argument in [98, Page 8].*

We are now in a position to describe the projection estimators for $f_\mu(\cdot)$ and $f_\nu(\cdot)$, and the rates achieved by the corresponding plug-in estimator.

**Theorem 2.5.** *Assume that $T_0(\cdot)$ is L-Lipschitz and $f_\mu$, $f_\nu$ are Lebesgue densities satisfying Assumption (A1). Also suppose that $K(\cdot)$ satisfies Assumption (A2). Define $h_m := m^{-\frac{1}{d+2s}} \log m$, $h_n := n^{-\frac{1}{d+2s}} \log n$ and $T := \int |K_d(u)|\, du + 1$. Fix any $M' > 0$. Consider any probability density $\widetilde{f}_\mu^{M'}(\cdot) \in C^s(\mathcal{X}; TM)$ (where $M$ is defined as in Assumption (A1)) which satisfies*

$$d_{\mathrm{IP}}\left(\widetilde{f}_\mu^{M'}, \widehat{f}_\mu; C^{s+2}(\mathcal{X}; M')\right) \leq \inf_{\substack{f(\cdot) \in C^s(\mathcal{X}; TM) \\ f \geq 0,\ \int f = 1}} d_{\mathrm{IP}}\left(\widehat{f}_\mu, f; C^{s+2}(\mathcal{X}; M')\right) + r_{d,s}^{(m,n)} \qquad (2.6)$$

*where $r_{d,s}^{(m,n)}$ is defined as in (2.7) and $d_{\mathrm{IP}}(\cdot, \cdot; C^{s+2}(\mathcal{X}; M'))$ is the integral probability metric defined in Definition 2.5. We define $\widetilde{f}_\nu^{M'}(\cdot)$ analogously as in (2.6) with $\mathcal{X}$, $\widehat{f}_\mu(\cdot)$ replaced by $\mathcal{Y}$, $\widehat{f}_\nu(\cdot)$. Then the following conclusions hold.*

1. *Set $M' := 8(1 + TM)$. If $\widetilde{\mu}_m$ and $\widetilde{\nu}_n$ are the probability measures corresponding to the probability densities $\widetilde{f}_\mu^{M'}(\cdot)$ and $\widetilde{f}_\nu^{M'}(\cdot)$, then the following holds for some constant $C > 0$:*

$$\mathbb{E}\left[\sup_{\gamma \in \widetilde{\Gamma}_{\min}} \int \|\widetilde{T}_{m,n}^\gamma(x) - T_0(x)\|^2\, d\widetilde{\mu}_m(x)\right] \leq C r_{d,s}^{(m,n)},$$

*where* $\quad r_{d,s}^{(m,n)} := \begin{cases} m^{-1/2} + n^{-1/2} & \text{for } d < 2(s+2), \\ m^{-1/2}\left(\log(1+m)\right)^d + n^{-1/2}\left(\log(1+n)\right)^d & \text{for } d = 2(s+2), \\ m^{-\frac{s+2}{d}} + n^{-\frac{s+2}{d}} & \text{for } d \geq 2(s+2). \end{cases}$ $\quad (2.7)$

*The same bound also holds for $\mathbb{E}|W_2^2(\widetilde{\mu}_m, \widetilde{\nu}_n) - W_2^2(\mu, \nu)|$.*

2. *$\widehat{f}_\mu(\cdot)$ satisfies*

$$\lim_{n \to \infty} \max\left\{\mathbb{P}\left(\|\widehat{f}_\mu\|_{C^s(\widetilde{\mathcal{X}})} \geq TM\right), \mathbb{P}\left(\sup_{x \in \widetilde{\mathcal{X}}} |\widehat{f}_\mu(x) - f_\mu(x)| \geq \varepsilon\right)\right\} = 0 \qquad (2.8)$$

*for any $\varepsilon > 0$, where $\widetilde{\mathcal{X}}$ is any compact subset of $\mathcal{X}^o$. The same conclusion holds for $\widehat{f}_\nu(\cdot)$ with $\mathcal{X}$ replaced by $\mathcal{Y}$.*

In Theorem 2.5, we have shown that the plug-in estimator for $T_0(\cdot)$ using $\widetilde{f}_\mu^{M'}(\cdot)$ and $\widetilde{f}_\nu^{M'}(\cdot)$ (with $M' = 8(1 + TM)$) achieves rates that mitigate the curse of dimensionality under sufficient smoothness. In fact, $\widetilde{f}_\mu^{M'}(\cdot)$ can be viewed as an approximate minimizer of $d_{\mathrm{IP}}(\widehat{f}_\mu, \cdot; C^{s+2}(\mathcal{X}, M'))$ over an appropriate class of Sobolev smooth probability densities. This is carried out because $\widehat{f}_\mu(\cdot)$ by itself may not be a probability density.

Further note that $\widetilde{\mu}_m, \widetilde{\nu}_n$ as specified in Theorem 2.5 are both smooth, and consequently $\widetilde{\Gamma}_{\min}$ is a singleton and the supremum in Theorem 2.5 can be dropped. A brief description of the proof technique for Theorem 2.5 is presented in Remark 2.6 below and the actual proof is given in Appendix C.1.

**Remark 2.6** (Proof technique). *The proof of Theorem 2.5 proceeds along the same lines as Remark 2.3. We first show that $\mathcal{F}_s$ (see Remark 2.2) can be chosen as a certain subset of $C^{s+2}(\mathcal{Y}^\circ)$. We then use Dudley's entropy integral type bounds which in turn requires the bracketing entropy [126, Page 270] of the class of compactly supported Sobolev smooth functions which can be found in [127, Corollary 2.7.2].*

We now explain the implications of both the parts of Theorem 2.5 in the following two remarks.

**Remark 2.7** (Mitigating the curse of dimensionality). *Theorem 2.5 shows that, under enough smoothness, i.e., when $2(s + 2) > d$, both the upper bounds for (1.8) and (1.9) are $O_p(n^{-1/2})$. This shows that, for large dimensions, provided $\mu$ and $\nu$ admit smooth enough densities, it is possible to construct plug-in estimators that mitigate the curse of dimensionality. Note that a similar estimator was analyzed in [67, Proposition 1] when $m = n$. However, the rates obtained in Theorem 2.5 are* strictly better *than those in [67, Proposition 1]. For $m = n$, when $d < 2(s + 2)$, [67] obtained a rate of $n^{-\frac{s+2}{2(s+2)+d}}$ which is* worse *than $n^{-1/2}$ obtained in Theorem 2.5. For the other regimes, [67] obtains rates (up to log factors) of $n^{-1/4}$ and $n^{-\frac{1}{(s+2)(d+2(s+2))}}$ which are both worse than the respective rates of $n^{-1/2}$ and $n^{-\frac{s+2}{d}}$ in Theorem 2.5.*

**Remark 2.8** (Computational aspects of Theorem 2.5). *Note that $\widetilde{f}_\mu^{M'}(\cdot)$ (with $M' = 8(1 + TM)$) is hard to compute whereas $\widehat{f}_\mu(\cdot)$ is computable easily in linear time. Note that if $\widehat{f}_\mu(\cdot)$ itself were a probability density in $C^s(\mathcal{X}; TM)$, then we would have $\widehat{f}_\mu = \widetilde{f}_\mu^{M'}$. While Theorem 2.5 does not establish that, it does come close in part 2, from which we can easily derive the following:*

$$\lim_{n \to \infty} \mathbb{P}(\widehat{f}_\mu(\cdot) \notin C^s(\widetilde{\mathcal{X}}; TM)) = 0.$$

*The above shows that $\widehat{f}_\mu(\cdot)$ is indeed bounded below by $(TM)^{-1}$ on $\widetilde{\mathcal{X}}$ (any compact subset of the interior of $\mathcal{X}$), and additionally belongs to $C^s(\widetilde{\mathcal{X}}; TM)$ with probability converging to 1. This leads us to conjecture that the natural density version of $\widehat{f}_\mu(\cdot)$, i.e.,*

$$\frac{\max\{\widehat{f}_\mu(\cdot), 0\}}{\int \max\{\widehat{f}_\mu(x), 0\} \, dx}$$

*should serve as a good proxy for $\widetilde{f}_\mu^{M'}(\cdot)$ and lead to rates of convergence that mitigate the curse of dimensionality. From a computational perspective, the density specified above is easy to simulate from using an accept-reject algorithm without computing the integral in the denominator (see [101, Algorithm 4.3]). However, our current proof technique does not provide rates of convergence for the above density estimator based on $\widehat{f}_\mu(\cdot)$.*

Another important implication of Theorem 2.5 is the bound obtained on $|W_2(\widetilde{\mu}_m, \widetilde{\nu}_n) - W_2(\mu, \nu)|$ when $\mu \neq \nu$. We first present the result and then describe the implication.

**Proposition 2.6.** *Consider the setting in Theorem 2.5. Then, provided $\mu \neq \nu$, the following holds:*

$$|W_2(\widetilde{\mu}_m, \widetilde{\nu}_n) - W_2(\mu, \nu)| = O_p(r_{d,s}^{(m,n)}).$$

Proposition 2.6 (see Appendix C.1 for a proof) shows an interesting distinction between the $\mu \neq \nu$ case and the $\mu = \nu$ case. For $\mu = \nu$, the best possible exponent is $n^{-\frac{1+s}{2s+d}}$ for $d \geq 3$ (see [131, Theorem 3] where the result was established under more general Besov smoothness assumptions). On the contrary, when $\mu \neq \nu$, Proposition 2.6 establishes a rate of $n^{-\frac{s+2}{d}}$ for the Wasserstein distance which is *strictly better* than the minimax achievable rate mentioned above when $\mu = \nu$. This observation complements [30, Corollary 1] where the authors make a similar remark for the special case of $s = 0$.

## 2.3 Discretized plug-in estimator under smoothness assumptions

In Section 2.1, we discussed how smoothness can be incorporated into the plug-in procedure to get faster rates of convergence. Such plug-in estimators are popular in the computational OT literature (see [7, 8, 25, 36]). However, even after $\widetilde{f}_\mu(\cdot) \equiv \widetilde{f}_\mu^{M'}(\cdot)$, $\widetilde{f}_\nu(\cdot) \equiv \widetilde{f}_\nu^{M'}(\cdot)$ are calculated, $\widetilde{T}_{m,n}^\gamma$ as in Theorem 2.5 cannot be computed explicitly from data if $\widetilde{f}_\mu(\cdot)$ and $\widetilde{f}_\nu(\cdot)$ are continuous densities. This is in contrast to $\widetilde{T}_{m,n}^\gamma$ from Theorem 2.2 in the *discrete-discrete* case which is explicitly computable using a standard linear program, but achieves worse rates of convergence. This is not unexpected. Thanks to the *no free lunch* principle, better statistical accuracy is naturally accompanied by heavier computational challenges. Therefore, our goal here is to construct estimators, under smoothness assumptions as in Section 2.2, which are computable in polynomial time (with complexity increasing with smoothness) provided $\widetilde{f}_\mu(\cdot)$ and $\widetilde{f}_\nu(\cdot)$ can be sampled from, and also attain rates that mitigate the curse of dimensionality.

*Construction*: We will illustrate the discretized estimator using the kernel based estimator from Section 2.2. Similar results also hold for the wavelet based estimator from Appendix A. Recall the kernel density estimators $\widetilde{f}_\mu(\cdot)$ and $\widetilde{f}_\nu(\cdot)$ (see (2.6)). Sample $M \geq 1$ random points from both $\widetilde{f}_\mu(\cdot)$ and $\widetilde{f}_\nu(\cdot)$. Let $\widehat{\mu}_{m,M}$ and $\widehat{\nu}_{n,M}$ denote the standard empirical measures on the $M$ points sampled from $\widetilde{f}_\mu(\cdot)$ and $\widetilde{f}_\nu(\cdot)$ respectively. Finally construct $\widetilde{T}_{m,n} \equiv \widetilde{T}_{m,n}^\gamma$ as in Definition 1.2 with $\widetilde{\mu}_m = \widehat{\mu}_{m,M}$ and $\widetilde{\nu}_n = \widehat{\nu}_{n,M}$. It should be pointed out that a similar construction was also used in [131, Section 6] for estimating probability densities under the Wasserstein loss. Based on this construction, the main result of this section is as follows:

**Theorem 2.7.** *Consider the setting in Theorem 2.5 and the same construction of $\widetilde{T}_{m,n}^\gamma$ as above. For simplicity, let's also assume $m = n$. Accordingly set $M = n^{\frac{s+2}{2}}$. Then $\widetilde{\Gamma}_{\min}$ is a singleton and consequently the following conclusion holds for some constant $C > 0$:*

$$\mathbb{E}\left[\int \|\widetilde{T}_{m,n}(x) - T_0(x)\|^2 \, d\widetilde{\mu}_m(x)\right] \leq C r_{d,s}^{(n,n)}.$$

*The same rates also hold for $\mathbb{E}|W_2^2(\widetilde{\mu}_m, \widetilde{\nu}_n) - W_2^2(\mu, \nu)|$.*

The proof of Theorem 2.7 is given in Appendix C.1. Once the empirical measures $\widehat{\mu}_{m,M}$ and $\widehat{\nu}_{n,M}$ have been obtained, an explicit computation of $\widetilde{T}_{m,n}$ as described above requires $O(M^3) = O(n^{\frac{3(s+2)}{2}})$ steps using the *Hungarian algorithm*, see [73]. This highlights the statistical versus computational trade-off, i.e., in order to mitigate the curse of dimensionality in convergence rates by exploiting smoothness, the computational complexity gets progressively worse by polynomial factors in $n$. It should be mentioned that (approximate) algorithms faster than the Hungarian algorithm stated above, can be found in [1, 36, 53] to name a few. Due to space constraints, we avoid a detailed discussion on this.

In the above construction, sampling from the smoothed kernel densities $\widetilde{f}_\mu(\cdot)$ and $\widetilde{f}_\nu(\cdot)$ is crucial. If we would simply draw $M$ bootstrap samples from the empirical distributions $\widehat{\mu}_m$ and $\widehat{\nu}_n$, the rates of convergence wouldn't improve from those observed in Theorem 2.2 no matter how large $M$ is.

## Acknowledgments and Disclosure of Funding

We would like to thank the reviewers for their constructive suggestions that greatly helped improve the quality of the paper. The third author is supported by NSF Grant DMS-2015376.

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
