# Supplement to "Rates of Estimation of Optimal Transport Maps using Plug-in Estimators via Barycentric Projections"

**Nabarun Deb**
Columbia University

**Promit Ghosal**
MIT

**Bodhisattva Sen**
Columbia University

In this file, we present the proofs of the main results, additional definitions and technical discussions that were deferred from the main paper.

## A  Wavelet based estimators

In this section, we will show how the curse of dimensionality in estimating $T_0(\cdot)$ can be mitigated under Besov type smoothness assumptions, by using wavelet based plug-in estimators for the probability densities associated with $\mu$ and $\nu$. We begin by defining the Besov class of functions which will play a pivotal role in the sequel.

**Definition A.1** (Besov class of functions). *We describe Besov classes following the notation from [131, Section 2.1.1]. Suppose $s > 0$ and let $n > s$ be a positive integer. Given $\Omega \subseteq \mathbb{R}^d$, $h \in \mathbb{R}^d$ and $f(\cdot) : \mathbb{R}^d \to \mathbb{R}^d$, set*

$$\mathbf{\Delta}_h^1 f(x) := f(x + h) - f(x),$$

$$\mathbf{\Delta}_h^k f(x) := \mathbf{\Delta}_h^1 \left( \mathbf{\Delta}_h^{k-1} f \right)(x), \quad \forall \, 2 \leq k \leq n,$$

*where these functions are defined on $\Omega_{h,n} := \{x \in \Omega : x + nh \in \Omega\}$. For $t > 0$, we then define*

$$\omega_n(f, t) := \sup_{\|h\| \leq t} \|\mathbf{\Delta}_h^n f\|_{L^2(\Omega_{h,n})}.$$

*Finally, we define the space $\mathcal{B}^s(\Omega)$ to be the set of functions for which the quantity*

$$\|f\|_{\mathcal{B}^s(\Omega)} := \|f\|_{L^2(\Omega)} + \sum_{j \geq 0} 2^{sj} \omega_n(f, 2^{-j})$$

*is finite. The above expression can also be used to define Besov spaces (and norms) for $s < 0$; see [33, Theorem 3.8.1].*

In this subsection, we assume that $\mu$ and $\nu$ admit Besov smooth densities $f_\mu(\cdot)$ and $f_\nu(\cdot)$ (see [33] and Definition A.1 above for details). Given $\Omega \subseteq \mathbb{R}^d$ and $s > 0$, let $\mathcal{B}^s(\Omega)$ denote the set of Besov smooth functions on $\Omega$ of order $s$.

**Assumption (A1)** (Regularity of the densities). *Suppose that:*

1. *$f_\mu$ and $f_\nu$ are supported on compact and convex subsets of $\mathbb{R}^d$, say $\mathcal{X}$ and $\mathcal{Y}$ respectively.*

2. *There exists $s, M > 0$ such that $\|f_\mu\|_{\mathcal{B}^s(\mathcal{X})} \leq M$, $\|f_\nu\|_{\mathcal{B}^s(\mathcal{Y})} \leq M$ and $f_\mu(x), f_\nu(y) \geq M^{-1}$ for all $x \in \mathcal{X}, y \in \mathcal{Y}$.*

We now present our wavelet based estimators for $f_\mu(\cdot)$ and $f_\nu(\cdot)$. Towards this direction, we begin with *sets* of functions in $L^2(\mathcal{X})$ (set of square integrable functions on $\mathcal{X}$), $\mathbf{\Phi}$ and $\{\mathbf{\Psi}_j\}_{j \geq 0}$, which form an orthonormal basis of $L^2(\mathcal{X})$ and satisfy the standard regularity assumptions for a wavelet basis (see [70, 95], [131, Appendix E]). We defer a formal discussion on these assumptions

35th Conference on Neural Information Processing Systems (NeurIPS 2021).

to Definition D.3 in the Appendix so as not to impede the flow of the paper. For the moment, it is worth noting that such sets of functions (e.g., Haar wavelets, Daubechies wavelets) are readily available in standard statistical softwares, see e.g., the R package `wavelets`.

Next, fix $J_m \in \mathbb{N}$ (a truncation parameter to be chosen later depending on the sample size $m$). Consider the following:

$$\widehat{f}_\mu(x) := \sum_{\phi \in \Phi} a_\phi \phi(x) + \sum_{j=0}^{J_m} \sum_{\psi \in \Psi_j} b_\psi \psi(x), \tag{A.1}$$

where

$$a_\phi := \frac{1}{m} \sum_{i=1}^m \phi(X_i), \qquad b_\psi := \frac{1}{m} \sum_{i=1}^m \psi(X_i).$$

Unfortunately $\widehat{f}_\mu(\cdot)$ as defined in (A.1) may not be a probability density and consequently cannot be used to obtain plug-in estimators for $\widetilde{T}_{m,n}^\gamma(\cdot)$. We therefore take the same route as in [131, Section 4.1] to define the following estimator for $f_\mu(\cdot)$:

$$\widetilde{f}_\mu := \min_{g \in \mathcal{D}(\mathcal{X})} \|g - \widehat{f}_\mu\|_{\mathcal{B}^{-1}(\mathcal{X})}, \tag{A.2}$$

where $\mathcal{D}(\mathcal{X})$ is the space of probability density functions on $\mathcal{X}$ and $\mathcal{B}^{-1}(\mathcal{X})$ is the Besov norm on $\mathcal{X}$ of order $-1$ as stated in Definition A.1. We can define $\widetilde{f}_\nu(\cdot)$ similarly. Computing both $\widetilde{f}_\mu(\cdot)$ and $\widehat{f}_\mu(\cdot)$ (as it involves infinite sums) is challenging and we would refer the interested reader to [131, Section 6] and the references therein, for details. Further discussion of this aspect is beyond the scope of this paper.

We are now in a position to present the main theorem of this subsection.

**Theorem A.1.** *Suppose that $T_0(\cdot)$ is $L$-Lipschitz, and $\widetilde{\mu}_m$ and $\widetilde{\nu}_n$ are the probability measures corresponding to the probability densities $\widetilde{f}_\mu(\cdot)$ and $\widetilde{f}_\nu(\cdot)$ with $m^{\frac{1}{d+2s}} \leq 2^{J_m} \leq m^{\frac{1}{d}}$ and $n^{\frac{1}{d+2s}} \leq 2^{J_n} \leq n^{\frac{1}{d}}$, then the following holds for some constant $C > 0$:*

$$\mathbb{E}\left[\sup_{\gamma \in \widetilde{\Gamma}_{\min}} \int \|\widetilde{T}_{m,n}^\gamma(x) - T_0(x)\|^2 \, d\widetilde{\mu}_m(x)\right] \leq C\widetilde{r}_{d,s}^{(m,n)}, \tag{A.3}$$

*where* $\quad \widetilde{r}_{d,s}^{(m,n)} := \begin{cases} m^{-1/2} \log(1+m) + n^{-1/2} \log(1+n) & \text{for } d = 2, \\ m^{-\frac{1+s}{d+2s}} + n^{-\frac{1+s}{d+2s}} & \text{for } d \geq 3, \end{cases}$ $\tag{A.4}$

*The same bound also holds for $\mathbb{E}|W_2^2(\widetilde{\mu}_m, \widetilde{\nu}_n) - W_2^2(\mu, \nu)|$.*

Note that $\frac{1+s}{d+2s} \to \frac{1}{2}$ as $s \to \infty$. Therefore Theorem A.1 shows that, when $m = n$, the rate of convergence for the wavelet based estimator is "close" to $n^{-1/2}$ provided $s$ is large enough for each fixed $d$. This shows that $T_0(\cdot)$ obtained using the wavelet estimators for $f_\mu(\cdot)$ and $f_\nu(\cdot)$ mitigates the curse of dimensionality, contrast this with the estimator in Theorem B.1. To avoid repetition, we defer further discussions on the rates observed in Theorem A.1 to Remark 2.7 where a holistic comparison is drawn with two other "smooth" plug-in estimators.

## B  Applications

In this section, we will apply our results to two popular problems, namely — estimating the Wasserstein barycenter between two probability distributions (see [2, 14, 24, 37]) in Appendix B.1, and obtaining detection thresholds in some recent optimal transport based independence testing procedures (see [6, 40, 55, 113, 114]) in Appendix B.2.

## B.1 Wasserstein barycenter estimation

Let $\mu, \nu \in \mathcal{P}_2(\mathbb{R}^d)$. The Wasserstein barycenter between $\mu$ and $\nu$ is then given by:

$$\rho_0 := \min_{\rho \in \mathcal{P}_{\mathrm{ac}}(\mathbb{R}^d)} \left( \frac{1}{2} W_2^2(\mu, \rho) + \frac{1}{2} W_2^2(\rho, \nu) \right). \tag{B.1}$$

In fact, by Proposition 1.1, there exists an optimal transport map $T_0$ from $\mu$ to $\nu$ and by [2, 14, 17], an alternative characterization of $\rho_0$ is as follows:

$$\rho_0 = \left( \frac{1}{2} \mathrm{Id} + \frac{1}{2} T_0 \right) \# \mu, \qquad \text{where} \qquad \mathrm{Id}(x) = x. \tag{B.2}$$

Estimating $\rho_0$ as in (B.1) has attracted significant attention over the past few years in economics [23, 28], Bayesian learning [119, 120], dynamic formulations [27, 32], algorithmic fairness [31, 62], etc. The most natural strategy employed in estimating $\rho_0$ is to use the empirical plug-in estimator, i.e., replacing $\mu, \nu$ in (B.1) with $\widehat{\mu}_m, \widehat{\nu}_n$. This strategy has been used, approximated and analyzed extensively in e.g., [17, 24, 37, 85]. Based on (B.2), the natural plug-in estimator of $\rho_0$ would be:

$$\widehat{\rho}_0^\gamma = \left( \frac{1}{2} \mathrm{Id} + \frac{1}{2} \widetilde{T}_{m,n}^\gamma \right) \# \widetilde{\mu}_m \tag{B.3}$$

where $\widetilde{T}_{m,n}^\gamma$ is the plug-in estimator of $T_0$ obtained by solving (1.7), with $\mu$ and $\nu$ replaced by $\widetilde{\mu}_m$ and $\widehat{\nu}_n$ respectively and $\gamma \in \widetilde{\Gamma}_{\min}$. While the consistency of $\widehat{\rho}_0^\gamma$ has been analyzed for $m = n$ in [85] and rates have been obtained for $d = 1$ in [13], the more general question of obtaining rates of convergence for $\widehat{\rho}_0^\gamma$ for general dimensions $d \geq 1$ is yet unanswered. We address this question in the following result (see Appendix C.2 for a proof).

**Theorem B.1.** *Suppose that the same assumptions from Theorem 2.2 hold. Then, with $\widehat{\rho}_0^\gamma$ as defined in (B.3) and $r_d^{(m,n)}$, $t_{d,\alpha}$ defined in Theorem 2.2, the following holds:*

$$\sup_{\gamma \in \widetilde{\Gamma}_{\min}} W_2^2(\widehat{\rho}_0^\gamma, \rho_0) = O_p \left( r_d^{(m,n)} \times (\log(1 + \max\{m, n\}))^{t_{d,\alpha}} \right).$$

## B.2 Nonparametric independence testing: Optimal transport based Hilbert-Schmidt independence criterion

Let $(X_1, Y_1), \ldots, (X_n, Y_n) \overset{i.i.d.}{\sim} \pi$, a probability measure on $\mathbb{R}^{d_1 + d_2}$, with marginals $\mu \in \mathcal{P}_{\mathrm{ac}}(\mathbb{R}^{d_1})$ and $\nu \in \mathcal{P}_{\mathrm{ac}}(\mathbb{R}^{d_2})$. Our problem of interest is the following hypothesis testing problem, given as:

$$\mathrm{H}_0 : \pi = \mu \otimes \nu \qquad \text{versus} \qquad \mathrm{H}_1 : \pi \neq \mu \otimes \nu. \tag{B.4}$$

This is the classical nonparametric independence testing problem which has received a lot of attention in the statistics and machine learning literature (see [11, 63, 71, 122], and [44, 74] for a review). In keeping with the overall theme of this paper, our focus here will be on a large class of OT based independence testing procedures, introduced first in [6] followed by recent developments in [40, 113, 114]. These tests bear resemblance to the Hilbert-Schmidt independence criterion (HSIC); see [63–65] and have attractive properties such as distribution-freeness (see Proposition B.2), consistency without moment assumptions and robustness against heavy-tailed distributions and against contamination [6, 113]. Below, we describe this class of tests, see (B.5) and (B.7). Our main theoretical contribution of this section will be to provide detection thresholds of these OT based tests.

*Construction*: Suppose $\upsilon_1, \upsilon_2$ be two compactly supported probability distributions on $\mathbb{R}^{d_1}$ and $\mathbb{R}^{d_2}$ respectively (e.g., $\upsilon_1 \equiv \mathrm{Unif}[0, 1]^{d_1}$, $\upsilon_2 \equiv \mathrm{Unif}[0, 1]^{d_2}$). Let $U_1, \ldots, U_n \overset{i.i.d.}{\sim} \upsilon_1$, $V_1, \ldots, V_n \overset{i.i.d.}{\sim} \upsilon_2$, $\widehat{u}_n := n^{-1} \sum_{i=1}^n \delta_{U_i}$ and $\widehat{v}_n := n^{-1} \sum_{j=1}^n \delta_{V_j}$. Recall the definitions of $\widehat{\mu}_n$ (with $m = n$) and $\widehat{\nu}_n$ from (1.2). Let $\widehat{T}_{1,n}$ ($\widehat{T}_{2,n}$) be obtained by solving (1.7), with $\mu$ and $\nu$ replaced by $\widehat{\mu}_n$ and $\widehat{u}_n$ ($\widehat{\nu}_n$ and $\widehat{v}_n$) respectively. Consider two non negative definite, continuous, characteristic kernels (see [52, 118] for definitions) $K_1(\cdot, \cdot)$ and $K_2(\cdot, \cdot)$ on $(\mathrm{supp}(\upsilon_1))^2$ and $(\mathrm{supp}(\upsilon_2))^2$. Set $\widehat{x}_{ij} := K_1(\widehat{T}_{1,n}(X_i), \widehat{T}_{1,n}(X_j))$ and $\widehat{y}_{ij} := K_2(\widehat{T}_{2,n}(Y_i), \widehat{T}_{2,n}(Y_j))$. Our test statistic is as follows:

$$\widehat{\mathrm{rHSIC}} := n^{-2} \sum_{i,j} \widehat{x}_{ij} \widehat{y}_{ij} + n^{-4} \sum_{i,j,r,s} \widehat{x}_{ij} \widehat{y}_{rs} - 2n^{-3} \sum_{i,j,r} \widehat{x}_{ij} \widehat{y}_{ir}. \tag{B.5}$$

**Proposition B.2** (See [6, 114])**.** *1.* ***(Distribution-freeness)*** *When $X_1$ and $Y_1$ are independent, the distribution of $n \times \widehat{\mathrm{rHSIC}}$ is universal, i.e., it does not depend on $\mu$ and $\nu$ for every fixed $n$.*

*2.* ***(Consistency against fixed alternatives)*** *Let $c_{n,\alpha}$ be the upper $(1 - \alpha)$-th quantile from the universal distribution in part 1 above. Then $\widehat{\mathrm{rHSIC}} \xrightarrow{P} \mathrm{rHSIC}(\pi | \mu \otimes \nu)$ where*

$$
\begin{aligned}
\mathrm{rHSIC}(\pi | \mu \otimes \nu) := & \, \mathbb{E}[K_1(T_1(X_1), T_1(X_2)) K_2(T_2(Y_1), T_2(Y_2))] + \mathbb{E}[K_1(T_1(X_1), T_1(X_2))] \\
& \times \mathbb{E}[K_2(T_2(Y_1), T_2(Y_2))] - 2\mathbb{E}[K_1(T_1(X_1), T_1(X_2)) K_2(T_2(Y_1), T_2(Y_3))],
\end{aligned}
\tag{B.6}
$$

*where $T_1(\cdot)$ (respectively $T_2(\cdot)$) is the optimal transport map from $\mu$ ($\nu$) to $\upsilon_1$ ($\upsilon_2$); see Definition 1.1. Further $\mathrm{rHSIC}(\pi | \mu \otimes \nu) = 0$ if and only if $\pi = \mu \otimes \nu$. Define the following test function:*

$$
\phi_{n,\alpha} := \mathbb{1}(n \times \widehat{\mathrm{rHSIC}} \geq c_{n,\alpha}).
\tag{B.7}
$$

*Then $\mathbb{E}[\phi_{n,\alpha}] \to 1$ as $n \to \infty$ under $H_1$, i.e., when $\pi \neq \mu \otimes \nu$.*

Proposition B.2 shows that the test based on $\widehat{\mathrm{rHSIC}}$ (see (B.5)), i.e., $\phi_{n,\alpha}$ (see (B.7)), can be carried out without resorting to the permutation principle as is necessary for the usual HSIC based test (see [64]). Further, when the sampling distribution is fixed, Proposition B.2 shows that $\widehat{\mathrm{rHSIC}}$ consistently estimates $\mathrm{rHSIC}(\pi | \mu \otimes \nu)$, a quantity which equals 0 if and only if $\pi = \mu \otimes \nu$ (this yields the consistency of $\phi_{n,\alpha}$) against fixed alternatives.

While consistency against fixed alternatives is an attractive feature of $\phi_{n,\alpha}$, a more intricate question of statistical interest is to understand the local power of $\phi_{n,\alpha}$ under "changing sequence of alternatives converging to the null" as $n \to \infty$. To study the local power of $\phi_{n,\alpha}$, we need to consider a triangular array setting, where the data distribution changes with $n$, i.e., $(X_1, Y_1), \ldots, (X_n, Y_n) \overset{i.i.d.}{\sim} \pi^{(n)}$, a probability measure on $\mathbb{R}^{d_1+d_2}$, with marginals $\mu^{(n)} \in \mathcal{P}_{\mathrm{ac}}(\mathbb{R}^{d_1})$ and $\nu^{(n)} \in \mathcal{P}_{\mathrm{ac}}(\mathbb{R}^{d_2})$. As $\mathrm{rHSIC}(\cdot | \cdot)$ characterizes independence, a mathematical formulation of "alternatives converging to null" would be to say $\mathrm{rHSIC}(\pi^{(n)} | \mu^{(n)} \otimes \nu^{(n)}) \to 0$ as $n \to \infty$. Similar questions have attracted a lot of attention in modern statistics, featuring measures (other than $\mathrm{rHSIC}(\cdot | \cdot)$) which characterize independence, see e.g., [5, 11, 79, 86]. In the following result (see Appendix C.2 for a proof), we show that if $\mathrm{rHSIC}(\pi^{(n)} | \mu^{(n)} \otimes \nu^{(n)}) \to 0$ slowly enough with $n$, then $\phi_{n,\alpha}$ yields a consistent sequence of tests for problem (B.4).

**Theorem B.3.** *Consider problem (B.4) with $\pi^{(n)}$, $\mu^{(n)}$, $\nu^{(n)}$ (changing with $n$) and suppose $T_{1,n}(\cdot)$ and $T_{2,n}(\cdot)$ are both $L$-Lipschitz ($L$ is free of $n$). Also assume $K_1(\cdot)$, $K_2(\cdot)$ are Lipschitz, $\mu^{(n)}$, $\nu^{(n)}$ are supported on fixed compact sets (supports are free of $n$). Set $r_{d_1,d_2}^{(n,n)} := r_{d_1}^{(n,n)} + r_{d_2}^{(n,n)}$ where $r_{d_1}^{(n,n)}, r_{d_2}^{(n,n)}$ is defined via (2.4). Then,*

$$
\mathbb{E}[\phi_{n,\alpha}] \to 1 \qquad \text{if} \qquad (r_{d_1,d_2}^{(n,n)})^{-1/2} \times \mathrm{rHSIC}(\pi^{(n)} | \mu^{(n)} \otimes \nu^{(n)}) \to \infty,
$$

## C  Proof of main results

This section is devoted to proving our main results and is organized as follows: In Appendix C.1, we present the proofs of results from Section 2 and in Appendix C.2, we present the proofs from Appendix B. Throughout this section, we will use the $\lesssim$ sign to hide constants that are free of $m, n$.

### C.1  Proofs from Section 2

*Proof of Theorem 2.1.* We begin the proof by observing that $\varphi_0^*(\cdot)$ is convex and finite on $\mathrm{supp}(\nu)$, and hence differentiable $\nu$ almost everywhere (a.e.). Further by Lemma D.2, we also have:

$$
\nabla \varphi_0^*(T_0(x)) = x \qquad \mu\text{-a.e. } x.
\tag{C.1}
$$

Fix any arbitrary $\gamma \in \widetilde{\Gamma}_{\min}$ and suppose that $\gamma(y|x)$ denotes the conditional distribution of $y$ given $x$ under $\gamma$. Define,

$$
D_1 := \int \varphi_0^*(y) \, d\widetilde{\nu}_n(y) - \int \varphi_0^*(y) d\nu_m^\dagger(y).
$$

As $\gamma$ has marginals $\widetilde{\mu}_m$ and $\widetilde{\nu}_n$, we have:

$$D_1 = \int_{x,y} \varphi_0^*(y)\, d\gamma(y|x)\, d\widetilde{\mu}_m(x) - \int_x \varphi_0^*(T_0(x))\, d\widetilde{\mu}_m(x). \qquad \text{(C.2)}$$

Next, by applying the conditional version of Jensen's inequality,

$$\int_x \left( \int_y \varphi_0^*(y)\, d\gamma(y|x) \right) d\widetilde{\mu}_m(x) \geq \int_x \varphi_0^* \left( \int_y y\, d\gamma(y|x) \right) d\widetilde{\mu}_m(x)$$

$$= \int_x \varphi_0^*(\widetilde{T}_{m,n}^\gamma(x))\, d\widetilde{\mu}_m(x). \qquad \text{(C.3)}$$

Using (C.3) with (C.2) yields,

$$D_1 \geq \int [\varphi_0^*(\widetilde{T}_{m,n}^\gamma(x)) - \varphi_0^*(T_0(x))] d\widetilde{\mu}_m(x)$$

$$\overset{(a)}{\geq} \int \left\{ \nabla \varphi_0^*(T_0(x))^\top (\widetilde{T}_{m,n}^\gamma(x) - T_0(x)) + \frac{1}{2L} \|\widetilde{T}_{m,n}^\gamma(x) - T_0(x)\|^2 \right\} d\widetilde{\mu}_m(x)$$

$$\overset{(b)}{=} \underbrace{\int x^\top (\widetilde{T}_{m,n}^\gamma(x) - T_0(x))\, d\widetilde{\mu}_m(x)}_{D_2} + \frac{1}{2L} \int \|\widetilde{T}_{m,n}^\gamma(x) - T_0(x)\|^2\, d\widetilde{\mu}_m(x). \qquad \text{(C.4)}$$

Here (a) follows from the strong convexity of $\varphi_0^*(\cdot)$ with parameter $(1/L)$ (see Lemma D.1) and (b) follows from (C.1).

Next, we will simplify the term $D_2$. Towards this direction, observe that for every $\gamma \in \widetilde{\Gamma}_{\min}$,

$$W_2^2(\widetilde{\mu}_m, \widetilde{\nu}_n) = \int \|x - y\|^2\, d\gamma(x,y)$$

$$= \int \|x\|^2\, d\widetilde{\mu}_m(x) + \int \|y\|^2\, d\widetilde{\nu}_n(y) - 2 \int_x \left( x^\top \int_y y\, d\gamma(y|x) \right) d\widetilde{\mu}_m(x)$$

$$= \int \|x\|^2\, d\widetilde{\mu}_m(x) + \int \|y\|^2\, d\widetilde{\nu}_n(y) - 2 \int_x x^\top \widetilde{T}_{m,n}^\gamma(x)\, d\widetilde{\mu}_m(x). \qquad \text{(C.5)}$$

Also, as $T_0$ is the gradient of a convex function, it is also an OT map from $\widetilde{\mu}_m$ to $\nu_m^\dagger$ (see [1, Section 1.2]), we have:

$$W_2^2(\widetilde{\mu}_m, \nu_m^\dagger) = \int \|x - T_0(x)\|^2\, d\widetilde{\mu}_m(x)$$

$$= \int \|x\|^2\, d\widetilde{\mu}_m(x) + \int \|y\|^2\, d\nu_m^\dagger(y) - 2 \int_x x^\top T_0(x)\, d\widetilde{\mu}_m(x). \qquad \text{(C.6)}$$

Now (C.5) and (C.6) imply

$$D_2 = \frac{1}{2} \left( W_2^2(\widetilde{\mu}_m, \nu_m^\dagger) - W_2^2(\widetilde{\mu}_m, \widetilde{\nu}_n) \right) + \frac{1}{2} \int \|y\|^2\, d(\widetilde{\nu}_n - \nu_m^\dagger)(y). \qquad \text{(C.7)}$$

Finally by combining (C.7) and (C.4), we get:

$$\frac{1}{2L} \int \|\widetilde{T}_{m,n}^\gamma(x) - T_0(x)\|^2\, d\widetilde{\mu}_m(x)$$

$$\leq \frac{1}{2} \left( W_2^2(\widetilde{\mu}_m, \widetilde{\nu}_n) - W_2^2(\widetilde{\mu}_m, \nu_m^\dagger) \right) + \int (\varphi_0^*(y) - (1/2)\|y\|^2)\, d(\widetilde{\nu}_n - \nu_m^\dagger)(y). \qquad \text{(C.8)}$$

Now note that the bound on the right hand side of the above display is free of the particular choice of $\gamma \in \widetilde{\Gamma}_{\min}$. Therefore, the same bound holds if we take a supremum over $\gamma \in \widetilde{\Gamma}_{\min}$ on the left hand side. We will now provide an upper bound for the right hand side of (C.8). The remainder of the proof proceeds as in the proof of [19, Proposition 2].

By the dual representation presented in (1.4) and (1.5), and the definitions of $\Psi_{\widetilde{\mu}_m, \widetilde{\nu}_n}(\cdot)$ and $\Psi_{\widetilde{\mu}_m, \nu_m^\dagger}(\cdot)$ in the statement of Theorem 2.1, we have

$$\frac{1}{2} W_2^2(\widetilde{\mu}_m, \widetilde{\nu}_n) = \frac{1}{2} \int \|x\|^2\, d\widetilde{\mu}_m(x) + \frac{1}{2} \int \|y\|^2\, d\widetilde{\nu}_n(y) - \mathcal{S}_{\widetilde{\mu}_m, \widetilde{\nu}_n}(\Psi_{\widetilde{\mu}_m, \widetilde{\nu}_n}),$$

and $\quad \frac{1}{2}W_2^2(\widetilde{\mu}_m, \nu_m^\dagger) = \frac{1}{2}\int \|x\|^2\, d\widetilde{\mu}_m(x) + \frac{1}{2}\int \|y\|^2\, d\nu_m^\dagger(y) - \mathcal{S}_{\widetilde{\mu}_m, \nu_m^\dagger}(\Psi_{\widetilde{\mu}_m, \nu_m^\dagger}).$

By subtracting the two equations above, we get:

$$\frac{1}{2}W_2^2(\widetilde{\mu}_m, \widetilde{\nu}_n) - \frac{1}{2}W_2^2(\widetilde{\mu}_m, \nu_m^\dagger) = \frac{1}{2}\int \|y\|^2\, d(\widetilde{\nu}_n - \nu_m^\dagger) - \mathcal{S}_{\widetilde{\mu}_m, \widetilde{\nu}_n}(\Psi_{\widetilde{\mu}_m, \widetilde{\nu}_n}) + \mathcal{S}_{\widetilde{\mu}_m, \nu_m^\dagger}(\Psi_{\widetilde{\mu}_m, \nu_m^\dagger}).$$
(C.9)

Next, we use (1.5) to make the following observations:

$$\mathcal{S}_{\widetilde{\mu}_m, \widetilde{\nu}_n}(\Psi_{\widetilde{\mu}_m, \widetilde{\nu}_n}) \leq \mathcal{S}_{\widetilde{\mu}_m, \widetilde{\nu}_n}(\Psi_{\widetilde{\mu}_m, \nu_m^\dagger}), \qquad \mathcal{S}_{\widetilde{\mu}_m, \nu_m^\dagger}(\Psi_{\widetilde{\mu}_m, \nu_m^\dagger}) \leq \mathcal{S}_{\widetilde{\mu}_m, \nu_m^\dagger}(\Psi_{\widetilde{\mu}_m, \widetilde{\nu}_n}). \qquad \text{(C.10)}$$

Note that (C.10) immediately yields the following conclusions:

$$\mathcal{S}_{\widetilde{\mu}_m, \nu_m^\dagger}(\Psi_{\widetilde{\mu}_m, \nu_m^\dagger}) - \mathcal{S}_{\widetilde{\mu}_m, \widetilde{\nu}_n}(\Psi_{\widetilde{\mu}_m, \nu_m^\dagger}) \leq \mathcal{S}_{\widetilde{\mu}_m, \nu_m^\dagger}(\Psi_{\widetilde{\mu}_m, \nu_m^\dagger}) - \mathcal{S}_{\widetilde{\mu}_m, \widetilde{\nu}_n}(\Psi_{\widetilde{\mu}_m, \widetilde{\nu}_n}),$$

and

$$\mathcal{S}_{\widetilde{\mu}_m, \nu_m^\dagger}(\Psi_{\widetilde{\mu}_m, \nu_m^\dagger}) - \mathcal{S}_{\widetilde{\mu}_m, \widetilde{\nu}_n}(\Psi_{\widetilde{\mu}_m, \widetilde{\nu}_n}) \leq \mathcal{S}_{\widetilde{\mu}_m, \nu_m^\dagger}(\Psi_{\widetilde{\mu}_m, \widetilde{\nu}_n}) - \mathcal{S}_{\widetilde{\mu}_m, \widetilde{\nu}_n}(\Psi_{\widetilde{\mu}_m, \widetilde{\nu}_n}).$$

By combining the above two displays, we have:

$$\left| \mathcal{S}_{\widetilde{\mu}_m, \nu_m^\dagger}(\Psi_{\widetilde{\mu}_m, \nu_m^\dagger}) - \mathcal{S}_{\widetilde{\mu}_m, \widetilde{\nu}_n}(\Psi_{\widetilde{\mu}_m, \widetilde{\nu}_n}) \right|$$
$$\leq \max \left\{ \left| \mathcal{S}_{\widetilde{\mu}_m, \nu_m^\dagger}(\Psi_{\widetilde{\mu}_m, \widetilde{\nu}_n}) - \mathcal{S}_{\widetilde{\mu}_m, \widetilde{\nu}_n}(\Psi_{\widetilde{\mu}_m, \widetilde{\nu}_n}) \right|, \left| \mathcal{S}_{\widetilde{\mu}_m, \nu_m^\dagger}(\Psi_{\widetilde{\mu}_m, \nu_m^\dagger}) - \mathcal{S}_{\widetilde{\mu}_m, \widetilde{\nu}_n}(\Psi_{\widetilde{\mu}_m, \nu_m^\dagger}) \right| \right\}.$$
(C.11)

By (1.5) and some simple algebra, the following holds:

$$\left| \mathcal{S}_{\widetilde{\mu}_m, \nu_m^\dagger}(\Psi_{\widetilde{\mu}_m, \widetilde{\nu}_n}) - \mathcal{S}_{\widetilde{\mu}_m, \widetilde{\nu}_n}(\Psi_{\widetilde{\mu}_m, \widetilde{\nu}_n}) \right| = \left| \int \Psi_{\widetilde{\mu}_m, \widetilde{\nu}_n}^*\, d(\nu_m^\dagger - \widetilde{\nu}_n) \right|.$$

A similar expression holds for $|\mathcal{S}_{\widetilde{\mu}_m, \nu_m^\dagger}(\Psi_{\widetilde{\mu}_m, \nu_m^\dagger}) - \mathcal{S}_{\widetilde{\mu}_m, \widetilde{\nu}_n}(\Psi_{\widetilde{\mu}_m, \nu_m^\dagger})|$. Using the above observation in (C.11), we get:

$$\left| \mathcal{S}_{\widetilde{\mu}_m, \nu_m^\dagger}(\Psi_{\widetilde{\mu}_m, \nu_m^\dagger}) - \mathcal{S}_{\widetilde{\mu}_m, \widetilde{\nu}_n}(\Psi_{\widetilde{\mu}_m, \widetilde{\nu}_n}) \right| \leq \max \left\{ \left| \int \Psi_{\widetilde{\mu}_m, \widetilde{\nu}_n}^*\, d(\widetilde{\nu}_n - \nu_m^\dagger) \right|, \left| \int \Psi_{\widetilde{\mu}_m, \nu_m^\dagger}^*\, d(\widetilde{\nu}_n - \nu_m^\dagger) \right| \right\}.$$

Combining the above display with (C.9), we further have:

$$\left| \frac{1}{2}W_2^2(\widetilde{\mu}_m, \widetilde{\nu}_n) - \frac{1}{2}W_2^2(\widetilde{\mu}_m, \nu_m^\dagger) - \left( \frac{1}{2}\int \|y\|^2\, d(\widetilde{\nu}_n - \nu_m^\dagger) \right) \right|$$
$$\leq \max \left\{ \left| \int \Psi_{\widetilde{\mu}_m, \widetilde{\nu}_n}^*\, d(\widetilde{\nu}_n - \nu_m^\dagger) \right|, \left| \int \Psi_{\widetilde{\mu}_m, \nu_m^\dagger}^*\, d(\widetilde{\nu}_n - \nu_m^\dagger) \right| \right\}. \qquad \text{(C.12)}$$

Combining (C.12) with (C.8) then completes the proof. $\qquad\qquad\qquad\qquad\qquad\square$

*Proof of Theorem 2.2.* First observe that

$$\limsup_{M \to \infty} \limsup_{m,n \to \infty} \mathbb{P}\left( \left| \int \varphi_0^*\, d(\widehat{\nu}_n - \nu_m^\dagger) \right| \geq M\left( r_d^{(m,m)} + r_d^{(n,n)} \right) \right) = 0$$

by the weak law of large numbers as $(r_d^{(n,n)})^{-1} n^{-1/2} = O(1)$ and $(r_d^{(m,m)})^{-1} m^{-1/2} = O(1)$.

Combining the above observation with Theorem 2.1, we have:

$$\limsup_{M \to \infty} \limsup_{m,n \to \infty} \mathbb{P}\left( \sup_{\gamma \in \widetilde{\Gamma}_{\min}} \int \|\widetilde{T}_{m,n}^\gamma(x) - T_0(x)\|^2\, d\widetilde{\mu}_m(x) \geq M r_d^{(m,n)} \right)$$
$$\leq \limsup_{M \to \infty} \limsup_{m,n \to \infty} \mathbb{P}\left( \max\left\{ \left| \int \Psi_{\widetilde{\mu}_m, \widetilde{\nu}_n}^*\, d(\widetilde{\nu}_n - \nu_m^\dagger) \right|, \left| \int \Psi_{\widetilde{\mu}_m, \nu_m^\dagger}^*\, d(\widetilde{\nu}_n - \nu_m^\dagger) \right| \right\} \geq \frac{M}{2} r_d^{(m,n)} \right)$$
$$\leq \limsup_{M \to \infty} \limsup_{m,n \to \infty} \left[ \mathbb{P}\left( \left| \int \Psi_{\widetilde{\mu}_m, \widetilde{\nu}_n}^*\, d(\widetilde{\nu}_n - \nu) \right| \geq \frac{M}{2} r_d^{(n,n)} \right) + \mathbb{P}\left( \left| \int \Psi_{\widetilde{\mu}_m, \widetilde{\nu}_n}^*\, d(\nu_m^\dagger - \nu) \right| \geq \frac{M}{2} r_d^{(m,m)} \right) \right.$$
$$\left. + \mathbb{P}\left( \left| \int \Psi_{\widetilde{\mu}_m, \nu_m^\dagger}^*\, d(\widetilde{\nu}_n - \nu) \right| \geq \frac{M}{2} r_d^{(n,n)} \right) + \mathbb{P}\left( \left| \int \Psi_{\widetilde{\mu}_m, \nu_m^\dagger}^*\, d(\nu_m^\dagger - \nu) \right| \geq \frac{M}{2} r_d^{(m,m)} \right) \right].$$
(C.13)

In the sequel, we will only discuss how to bound the first term on the right hand side of (C.13). Once that is understood, the other terms can be bounded similarly. Therefore, our focus is on showing

$$\limsup_{M\to\infty}\limsup_{m,n\to\infty}\mathbb{P}\left(\left|\int\Psi^*_{\widetilde{\mu}_m,\widetilde{\nu}_n}\,d(\widetilde{\nu}_n-\nu)\right|\geq\frac{M}{2}r_d^{(n,n)}(\log{(1+\max\{m,n\})})^{t_{d,\alpha}}\right)=0.\quad\text{(C.14)}$$

For the next part, to simplify notation, let us begin with some notation. Set $\mathcal{Y}:=\operatorname{supp}(\nu)$ and $\mathcal{X}_{n,\mu}$ denote the closure of the convex hull of $X_1,\ldots,X_n$.

Note that if we replace $\Psi_{\widetilde{\mu}_m,\widetilde{\nu}_n}(\cdot)$ by $\Psi_{\widetilde{\mu}_m,\widetilde{\nu}_n}(\cdot)-C$ for some constant $C>0$, then $\Psi^*_{\widetilde{\mu}_m,\widetilde{\nu}_n}(\cdot)\mapsto\Psi^*_{\widetilde{\mu}_m,\widetilde{\nu}_n}+C$. However replacing $\Psi^*_{\widetilde{\mu}_m,\widetilde{\nu}_n}(\cdot)$ by $\Psi^*_{\widetilde{\mu}_m,\widetilde{\nu}_n}(\cdot)+C$ in (C.14) doesn't change its value as $\widetilde{\nu}_n$ and $\nu$ are both probability measures. Therefore, without loss of generality, we can assume that $\Psi_{\widetilde{\mu}_m,\widetilde{\nu}_n}(X_1)=0$ for all $m,n$. We will stick to this convention for the rest of the proof. Also note that $\Psi_{\widetilde{\mu}_m,\widetilde{\nu}_n}(\cdot)$ is only determined at the data points $X_1,\ldots,X_n$. Without loss of generality, we extend $\Psi_{\widetilde{\mu}_m,\widetilde{\nu}_n}(\cdot)$ to the whole of $\mathbb{R}^d$ by linear interpolation for any $x\in\mathcal{X}_{n,\mu}$ and setting $\Psi_{\widetilde{\mu}_m,\widetilde{\nu}_n}(x)=\infty$ for $x\in\mathcal{X}^c_{n,\mu}$.

The proof now proceeds using the following steps:

**Step I:** There exists a constant $C_1>0$ and $y_n\in\operatorname{supp}(\nu)=\mathcal{Y}$ such that

$$|\Psi^*_{\widetilde{\mu}_m,\widetilde{\nu}_n}(y_n)|\leq\max_{1\leq i\leq m}\|X_i\|.$$

*Proof of step I.* By Kantorovich duality, there exists $y_n$ such that

$$\Psi^*_{\widetilde{\mu}_m,\widetilde{\nu}_n}(y_n)+\Psi_{\widetilde{\mu}_m,\widetilde{\nu}_n}(X_1)=\langle X_1,y_n\rangle\quad\implies\quad|\Psi^*_{\widetilde{\mu}_m,\widetilde{\nu}_n}(y_n)|\leq C_1\|X_1\|\leq C_1\max_{1\leq i\leq m}\|X_i\|,$$

where $C_1:=\sup\{\|y\|:\,y\in\mathcal{Y}\}$. $\qquad\square$

**Step II:** There exists a constant $C_2>0$ such that the following holds:

$$\|\Psi^*_{\widetilde{\mu}_m,\widetilde{\nu}_n}\|_{\infty,\mathcal{Y}}\leq C_2\max_{1\leq i\leq n}\|X_i\|,$$

where $\|\cdot\|_{\infty,\mathcal{Y}}$ is the uniform norm on the support of $\nu$.

*Proof of step II.* As $\Psi_{\widetilde{\mu}_m,\widetilde{\nu}_n}(x)=\infty$ for $x\in\mathcal{X}^c_{n,\mu}$, using (1.6), we can write $\Psi^*_{\widetilde{\mu}_m,\widetilde{\nu}_n}(y)=\max_{x\in\mathcal{X}_{n,\mu}}(\langle x,y\rangle-\Psi_{\widetilde{\mu}_m,\widetilde{\nu}_n}(x))$ for all $y\in\mathcal{Y}$. For any $y_0\in\mathcal{Y}$, let $x_0\in\mathcal{X}_{n,\mu}$ be such that $\Psi^*_{\widetilde{\mu}_m,\widetilde{\nu}_n}(y_0)=\langle x_0,y_0\rangle-\Psi_{\widetilde{\mu}_m,\widetilde{\nu}_n}(x_0)$. Then, for any $y\in\mathcal{X}$, we have:

$$\begin{cases}\Psi^*_{\widetilde{\mu}_m,\widetilde{\nu}_n}(y_0)=\langle x_0,y_0\rangle-\Psi_{\widetilde{\mu}_m,\widetilde{\nu}_n}(x_0)\\\Psi^*_{\widetilde{\mu}_m,\widetilde{\nu}_n}(y)\geq\langle x_0,y\rangle-\Psi_{\widetilde{\mu}_m,\widetilde{\nu}_n}(x_0)\end{cases}$$

$$\implies|\Psi^*_{\widetilde{\mu}_m,\widetilde{\nu}_n}(y_0)-\Psi^*_{\widetilde{\mu}_m,\widetilde{\nu}_n}(y)|\leq|\langle x_0,y_0-y\rangle|\leq\left(\max_{1\leq i\leq m}\|X_i\|\right)\|y_0-y\|.$$

where the last line uses the fact that $y_0,y$ are arbitrary. In particular, by setting $y_0:=y_n$ from step I, we get:

$$\|\Psi^*_{\widetilde{\mu}_m,\widetilde{\nu}_n}\|_{\infty,\mathcal{Y}}\leq|\Psi^*_{\widetilde{\mu}_m,\widetilde{\nu}_n}(y_n)|+\left(\max_{1\leq i\leq m}\|X_i\|\right)\sup_{y\in\mathcal{Y}}\|y_n-y\|\leq C_2\left(\max_{1\leq i\leq m}\|X_i\|\right),$$

where $C_2:=3C_1$ with $C_1$ defined as specified in the proof of step I. $\qquad\square$

The above lemma allows us to bound (with high probability) the $L^\infty$-norm of $\Psi^*_{\widetilde{\mu}_m,\widetilde{\nu}_n}(\cdot)$ on $\mathcal{Y}$, using the tail assumption $\mathbb{E}\exp(t\|X_1\|^\alpha)<\infty$ for some $t>0$ and $\alpha>0$. This is the focus of the next step.

**Step III**: For $K>0$, define the following two sets:

$$A_{m,n,K}:=\left\{\int(\Psi^*_{\widetilde{\mu}_m,\widetilde{\nu}_n}(u))^2\,d\nu(u)\geq K\right\},\quad\text{and,}$$

$$\widetilde{A}_{m,n,K} := \left\{ \|\Psi^*_{\widetilde{\mu}_m,\widetilde{\nu}_n}\|_{\infty,\mathcal{Y}} \geq K \left( \log n \right)^{1/\alpha} \right\}.$$

Then there exists $K_0 > 0$ such that for any $K \geq K_0$, we have:

$$\lim_{m,n \to \infty} \mathbb{P}(\widetilde{A}_{m,n,K}) = 0. \tag{C.15}$$

and

$$\lim_{m,n \to \infty} \mathbb{P}(A_{m,n,K}) = 0. \tag{C.16}$$

*Proof of step III.* By using the exponential Markov's inequality coupled with the standard union bound, we have:

$$\mathbb{P}\left( \max_{1 \leq i \leq m} \|X_i\| \geq K(\log m)^{1/\alpha} \right) \leq m \mathbb{P}\left( \|X_1\| \geq K(\log m)^{1/\alpha} \right)$$

$$\leq m \exp(-tK^\alpha(\log m)) \mathbb{E} \exp(t\|X_1\|^\alpha) \stackrel{m \to \infty}{\longrightarrow} 0$$

provided $K > t^{-\alpha}$. Using the above observation coupled with step II, (C.15) follows by choosing $K_0 > C_2 t^{-\alpha}$.

For the next part, we define another set:

$$B_{m,n,\varepsilon} := \left\{ \int \left| \Psi^*_{\widetilde{\mu}_m,\widetilde{\nu}_n}(u) - \Psi^*_{\mu,\nu}(u) \right|^2 d\nu(u) \geq \varepsilon \right\}$$

for $\varepsilon > 0$, where, as in (1.5), we have:

$$W_2^2(\mu,\nu) = \int \|x\|^2 d\mu(x) + \int \|y\|^2 d\nu(y) - 2 \left( \int \Psi_{\mu,\nu}(x) d\mu(x) + \int \Psi^*_{\mu,\nu}(y) d\nu(y) \right).$$

Now by using [7, Theorem 2.10], we have $\mathbb{P}(B_{m,n,\varepsilon}) \to 0$ as $m, n \to \infty$ for all $\varepsilon > 0$. As

$$\int (\Psi^*_{\widetilde{\mu}_m,\widetilde{\nu}_n}(u))^2 d\nu(u) \leq 2 \int \left| \Psi^*_{\widetilde{\mu}_m,\widetilde{\nu}_n}(u) - \Psi^*_{\mu,\nu}(u) \right|^2 d\nu(u) + 2 \int (\Psi^*_{\mu,\nu}(u))^2 d\nu(u),$$

(C.16) follows with $K_0 > 2 \int (\Psi^*_{\mu,\nu}(u))^2 d\nu(u) + 1$ if we choose $\epsilon = 1/2$. $\qquad\square$

We are now in a position to complete the proof of Theorem 2.2 using steps I-III. Towards this direction, set $K' := 2K_0$ where $K_0$ is defined as in the proof of step III and observe that for any $M > 0$,

$$\limsup_{M \to \infty} \limsup_{m,n \to \infty} \mathbb{P}\left( \left| \int \Psi^*_{\widetilde{\mu}_m,\widetilde{\nu}_n}(u) d(\widetilde{\nu}_n - \nu) \right| \geq M r_d^{(n,n)} (\log(1+m))^{t_{d,\alpha}} \right)$$

$$\leq \limsup_{M \to \infty} \limsup_{m,n \to \infty} \mathbb{P}\left( \left| \int \Psi^*_{\widetilde{\mu}_m,\widetilde{\nu}_n}(u) d(\widetilde{\nu}_n - \nu) \right| \geq M r_d^{(n,n)} (\log(1+m))^{t_{d,\alpha}}, A^c_{m,n,K'} \cap \widetilde{A}^c_{m,n,K'} \right)$$

$$+ \limsup_{n \to \infty} \mathbb{P}(\widetilde{A}_{m,n,K'}) + \limsup_{m,n \to \infty} \mathbb{P}(A_{m,n,K'})$$

$$\leq \limsup_{M \to \infty} \limsup_{m,n \to \infty} \mathbb{P}\left( \left| \int \Psi^*_{\widetilde{\mu}_m,\widetilde{\nu}_n}(u) d(\widetilde{\nu}_n - \nu) \right| \geq M r_d^{(n,n)} (\log(1+m))^{t_{d,\alpha}}, A^c_{m,n,K'} \cap \widetilde{A}^c_{m,n,K'} \right),$$
$$\tag{C.17}$$

where the last step follows from step III. Observe that the left hand side of (C.17) is the same as (C.14). Therefore, it is now enough to bound the right hand side of (C.17).

In order to achieve the above task, let us define the following class of functions:

$$\mathcal{C}^{\Gamma,L}(\mathcal{Y}) := \{ f : \mathcal{Y} \to \mathbb{R}, f \text{ is convex}, \|f\|_{\infty,\mathcal{Y}} \leq \Gamma, \|f\|_{L^2(\nu)} \leq L \}.$$

By setting $\Gamma := K'(\log m)^{1/\alpha}$ and $L := K'$, (C.17) yields the following conclusion:

$$\limsup_{M \to \infty} \limsup_{m,n \to \infty} \mathbb{P}\left( \left| \int \Psi^*_{\widetilde{\mu}_m,\widetilde{\nu}_n}(u) d(\widetilde{\nu}_n - \nu) \right| \geq M r_d^{(n,n)} (\log(1+m))^{t_{d,\alpha}} \right)$$

$$\leq \limsup_{M \to \infty} \limsup_{m,n \to \infty} \mathbb{P}\left( \sup_{f \in \mathcal{C}^{\Gamma,L}(\mathcal{Y})} \left| \int f \, d(\widetilde{\nu}_n - \nu) \right| \geq M r_d^{(n,n)} (\log(1+m))^{t_{d,\alpha}} \right).$$

By an application of Markov's inequality, it thus suffices to show that:

$$\mathbb{E}\left[\sup_{f \in \mathcal{C}^{\Gamma, L}(\mathcal{Y})} \left| \int f \, d(\widetilde{\nu}_n - \nu) \right|\right] = \mathcal{O}\left(r_d^{(n,n)}(\log(1+m))^{t_{d,\alpha}}\right). \tag{C.18}$$

In order to bound (C.18), we will use some standard empirical process techniques. In particular, by using [22, Theorem 5.11], the following bound holds:

$$\mathbb{E}\left[\sup_{f \in \mathcal{C}^{\Gamma, L}(\mathcal{Y})} \left| \int f \, d(\widetilde{\nu}_n - \nu) \right|\right]$$

$$\leq D \inf\left\{ a \geq \frac{\Gamma}{\sqrt{n}} : a \geq \frac{D}{\sqrt{n}} \int_a^\Gamma \sqrt{\log N_{[]}(\varepsilon, \mathcal{C}^{\Gamma, L}(\mathcal{Y}), L^2(\nu))} \, d\varepsilon \right\}, \tag{C.19}$$

for some positive constant $D > 0$, where $N_{[]}(\varepsilon, \mathcal{C}^{\Gamma, L}(\mathcal{Y}), L^2(\nu))$ is the $\varepsilon$-bracketing number of the class of functions $\mathcal{C}^{\Gamma, L}(\mathcal{Y})$ with respect to the $L^2(\nu)$ norm. Note that by [17, Equation 26], we have:

$$\log N_{[]}(\varepsilon, \mathcal{C}^{\Gamma, L}(\mathcal{Y}), L^2(\nu)) \leq \gamma_d \left(\log \frac{\Gamma}{\varepsilon}\right)^{d+1} \left(\frac{L}{\varepsilon}\right)^{d/2}$$

for some $\gamma_d > 0$ depending only on fand the diameter of $\mathcal{Y}$.

We will now bound the right hand side of (C.19). Also we will use $D_d$ to denote changing constants which can depend on $d$.

1. *When $d = 1, 2, 3$:* Choose $a = D_d \frac{(\log n)^{\frac{1}{\alpha} \vee \frac{2\alpha + 2d\alpha - d + 4}{4\alpha}}}{\sqrt{n}}$. Observe that:

$$\frac{1}{\sqrt{n}} \int_a^\Gamma \sqrt{\log N_{[]}(\varepsilon, \mathcal{C}^{\Gamma, L}(\mathcal{Y}), L^2(\nu))} \, d\varepsilon \leq \frac{(\log n)^{(d+1)/2}}{\sqrt{n}} \cdot \left[\frac{\varepsilon^{1-d/4}}{1 - d/4}\right]_0^\Gamma$$

$$\lesssim \frac{(\log n)^{(4-d)/(4\alpha)} \times (\log n)^{(d+1)/2}}{\sqrt{n}} \lesssim a.$$

2. *When $d = 4$:* Choose $a = D_d \frac{(\log n)^{\frac{1}{\alpha} \vee \frac{7}{2}}}{\sqrt{n}}$. Observe that:

$$\frac{1}{\sqrt{n}} \int_a^\Gamma \sqrt{\log N_{[]}(\varepsilon, \mathcal{C}^{\Gamma, L}(\mathcal{Y}), L^2(\nu))} \, d\varepsilon \leq \frac{(\log n)^{5/2}}{\sqrt{n}} \cdot [\log \varepsilon]_{D_d\Gamma/\sqrt{n}}^\Gamma$$

$$\lesssim \frac{(\log n)^{(7/2)}}{\sqrt{n}} \lesssim a.$$

3. *When $d > 4$:* Choose $a = D_d \frac{(\log n)^{2(1+d^{-1})}}{n^{2/d}}$. Observe that:

$$\frac{1}{\sqrt{n}} \int_a^{\Gamma_0} \sqrt{\log N_{[]}(\varepsilon, \mathcal{C}^{\Gamma, L}(\mathcal{Y}), L^2(\nu))} \, d\varepsilon \leq \frac{(\log n)^{(d+1)/2}}{\sqrt{n}} \cdot \left[\frac{\varepsilon^{1-d/4}}{1 - d/4}\right]_a^\Gamma$$

$$\lesssim \frac{a^{1-d/4}(\log n)^{(d+1)/2}}{\sqrt{n}} \lesssim a.$$

This completes the proof after applying the same technique on the other 3 terms on the right hand side of (C.13). □

*Proof of Corollary 2.3.* First observe that

$$\mathbb{E}\left[\int \varphi_0^* \, d\widehat{\nu}_n\right] = \mathbb{E}\left[\int \varphi_0^* \, d\nu_m^\dagger\right] = \int \varphi_0^* \, d\nu.$$

Using the above observation and the same approach used as in the proof of Theorem 2.2, we will only focus on bounding

$$\mathbb{E}\left| \int \Psi_{\widetilde{\mu}_m, \widetilde{\nu}_n}^* \, d(\widetilde{\nu}_n - \nu) \right|. \tag{C.20}$$

The general strategy to bound the term in (C.20) is derived from some intermediate steps in the proofs of [5, Lemmas 3 and 4]. We still present a sketch here for completeness.

By the same argument as in the proof of Theorem 2.2 and using the fact that there exists fixed $R > 0$ such that $\max_{1 \leq i \leq m} \|X_i\| \leq R$, we have $\Psi^*_{\widetilde{\mu}_m, \widetilde{\nu}_n}(\cdot)$ is a convex and $R$-Lipschitz function on $\mathcal{Y}$. This observation implies:

$$\mathbb{E}\left| \int \Psi^*_{\widetilde{\mu}_m, \widetilde{\nu}_n} \, d(\widetilde{\nu}_n - \nu_m^\dagger) \right| \leq \mathbb{E}\left[ \sup_{\psi \in \mathcal{F}_R(\mathcal{Y})} \left| \int \psi \, d(\widehat{\nu}_n - \nu) \right| \right] \tag{C.21}$$

where $\mathcal{F}_R(\mathcal{Y})$ is the set of convex and $R$-Lipschitz functions on $\mathcal{Y}$. By [25, Theorem 5.22], we then have:

$$\mathbb{E}\left[ \sup_{\psi \in \mathcal{F}_R(\mathcal{Y})} \left| \int \psi \, d(\widehat{\nu}_n - \nu) \right| \right] \lesssim \inf_{\delta > 0} \left( \delta + n^{-1/2} \int_\delta^{R^2} \sqrt{\log \mathcal{N}_\infty(\mathcal{F}_R(\mathcal{Y}), \varepsilon)} \, d\varepsilon \right), \tag{C.22}$$

where $\mathcal{N}_\infty(\mathcal{F}_R(\mathcal{Y}), \varepsilon)$ is the $\varepsilon$-covering number of the set $\mathcal{F}_R(\mathcal{Y})$ with respect to the uniform metric. By using [13, Theorem 1] (also see [4]), there exists constants $C_1, C_2 > 0$ such that whenever $\varepsilon/R^2 \leq C_1$, then $\log \mathcal{N}_\infty(\mathcal{F}_R(\mathcal{Y}), \varepsilon) \leq C_2(u/R^2)^{-d/2}$. By using this bound in (C.22), we get:

$$\mathbb{E}\left[ \sup_{\psi \in \mathcal{F}_R(\mathcal{Y})} \left| \int \psi \, d(\widehat{\nu}_n - \nu) \right| \right] \lesssim \inf_{\delta > 0} \left( \delta + n^{-1/2} \int_\delta^1 \varepsilon^{-d/4} \, d\varepsilon \right). \tag{C.23}$$

Setting $\delta = 0$ for $d < 4$ and $\delta = n^{-2/d}$ for $d \geq 4$ in (C.22), followed by a direct application of (C.21), we have:

$$\mathbb{E}\left| \int \Psi^*_{\widetilde{\mu}_m, \widetilde{\nu}_n} \, d(\widetilde{\nu}_n - \nu_m^\dagger) \right| \lesssim r_d^{(n,n)}.$$

This completes the proof. $\qquad \square$

*Proof of Theorem A.1.* For this proof, we will use an intermediate step in the proof of Theorem 2.1, which is (C.7), that can alternatively be written as:

$$\mathbb{E}\left[ \int \|\widetilde{T}_{m,n}^\gamma(x) - T_0(x)\|^2 \, d\widetilde{\mu}_m(x) \right] \lesssim \mathbb{E}|W_2^2(\widetilde{\mu}_m, \widetilde{\nu}_n) - W_2^2(\mu, \nu)| + \mathbb{E}\left| \int h(y) \, d(\widetilde{\nu}_n - \nu_m^\dagger)(y) \right| \tag{C.24}$$

where $h(y) := \varphi_0^*(y) - (1/2)\|y\|^2$ and $C > 0$ is some constant. As $\mathcal{X}$ and $\mathcal{Y}$ are compact sets, the function $h(\cdot)$ is Lipschitz. Therefore,

$$\mathbb{E}\left| \int h(y) \, d(\widetilde{\nu}_n - \nu)(y) \right| \lesssim W_1(\widetilde{\nu}_n, \nu) \leq W_2(\widetilde{\nu}_n, \nu).$$

Further, as $T_0(\cdot)$ is also Lipschitz, we further have:

$$\mathbb{E}\left| \int h(y) \, d(\nu_m^\dagger - \nu)(y) \right| \lesssim W_1(T_0 \# \widetilde{\mu}_m, T_0 \# \mu) \lesssim W_1(\widetilde{\mu}_m, \mu) \leq W_2(\widetilde{\mu}_m, \mu).$$

Finally, by the triangle inequality, we also have:

$$\mathbb{E}|W_2^2(\widetilde{\mu}_m, \widetilde{\nu}_n) - W_2^2(\mu, \nu)| \lesssim \mathbb{E}|W_2(\widetilde{\mu}_m, \widetilde{\nu}_n) - W_2(\widetilde{\mu}_m, \nu)| + \mathbb{E}|W_2(\widetilde{\mu}_m, \nu) - W_2(\mu, \nu)|$$
$$\leq \mathbb{E}W_2(\widetilde{\mu}_m, \mu) + \mathbb{E}W_2(\widetilde{\nu}_n, \nu).$$

Combining the three displays above and plugging them back in (C.24), we get:

$$\mathbb{E}\left[ \int \|\widetilde{T}_{m,n}^\gamma(x) - T_0(x)\|^2 \, d\widetilde{\mu}_m(x) \right] \lesssim \mathbb{E}W_2(\widetilde{\mu}_m, \mu) + \mathbb{E}W_2(\widetilde{\nu}_n, \nu).$$

The conclusion then follows from [131, Theorem 1]. $\qquad \square$

*Proof of Theorem 2.5.* **Part 1.** By the same arguments (see e.g., (C.13)) as used in the proof of Theorem 2.2, it suffices to show that

$$\mathbb{E}\left| \int \Psi^*_{\widetilde{\mu}_m, \widetilde{\nu}_n}(u)(\widetilde{f}_\nu^{M'}(u) - f_\nu(u)) \, du \right| \lesssim r_{d,s}^{(n,n)} \tag{C.25}$$

for some $M' > 0$.

The general structure of the proof is similar to that of Theorem 2.2. The crucial observation is that $\widetilde{f}_\mu^{M'}(\cdot)$ and $\widetilde{f}_\nu^{M'}(\cdot)$ are elements of $C^s(\mathcal{X}; TM)$ and $C^s(\mathcal{Y}; TM)$ respectively, for any $M' > 0$. Note that, by Caffarelli regularity theory; see [16, Theorem 33], there exists $M' > 0$ such that $\|\Psi^*_{\widetilde{\mu}_m, \widetilde{\nu}_n}(\cdot)\|_{C^{s+2}(\mathcal{Y})} \leq M'$.

Next, let us define the following class of functions:

$$\mathcal{G}_t^L(\mathcal{Y}) := \{g : \mathcal{Y} \to \mathbb{R}, \ g(\cdot) \text{ is convex, } \|g\|_{C^t(\mathcal{Y})} \leq L\}.$$

Observe that

$$\mathbb{E}\left|\int \Psi^*_{\widetilde{\mu}_m, \widetilde{\nu}_n}(u)(\widetilde{f}_\nu^{M'}(u) - f_\nu(u))\,du\right| \leq \mathbb{E}\sup_{g \in \mathcal{G}_{s+2}^{M'}(\mathcal{Y})}\left|\int g(u)(\widetilde{f}_\nu^{M'}(u) - f_\nu(u))\,du\right|$$

$$\leq 2\mathbb{E}\sup_{g \in \mathcal{G}_{s+2}^{M'}(\mathcal{Y})}\left|\int g(u)(\widehat{f}_\nu(u) - f_\nu(u))\,du\right| + r_{d,s}^{(n,n)}$$

(C.26)

where the last line follows from (2.6).

Set $K_{d,h_n}(\cdot) := h_n^{-d}K_d(\cdot/h_n)$. Following the same decomposition as in [21], we write:

$$\mathbb{E}\sup_{g \in \mathcal{G}_{s+2}^{M'}(\mathcal{Y})}\left|\int g(u)(\widehat{f}_\nu(u) - f_\nu(u))\,du\right|$$

$$= \mathbb{E}\sup_{g \in \mathcal{G}_{s+2}^{M'}(\mathcal{Y})}\left|\int g(u + u')K_{d,h_n}(u')\,d\widehat{\nu}_n(u)\,du' - \int g(u)f_\nu(u)\,du\right|$$

$$\leq \mathbb{E}\sup_{g \in \mathcal{G}_{s+2}^{M'}(\mathcal{Y})}\left|\int g(u + u')K_{d,h_n}(u')\,d(\widehat{\nu}_n - \nu)(u)\,du'\right|$$

$$+ \sup_{g \in \mathcal{G}_{s+2}^{M'}(\mathcal{Y})}\left|\int g(u + u')K_{d,h_n}(u')f_\nu(u)\,du\,du' - \int g(u)f_\nu(u)\,du\right|. \qquad (C.27)$$

We will now bound the two terms on the right hand side of (C.27). For the first term, define

$$\overline{g}_n(u) := \int g(u + u')K_{d,h_n}(u)\,du'.$$

If $g \in \mathcal{G}_t^L(\mathcal{Y}^o)$, then by [9, Proposition 8.10] and using Assumption (A2), we have $\overline{g}_n \in \mathcal{G}_{s+2}^{cM'}(\mathcal{Y}^o)$, for some constant $c > 0$ (depending on the constants involved in Assumption (A2) and the diameter of $\mathcal{Y}$). Combining these observations with (C.27), we get:

$$\mathbb{E}\sup_{g \in \mathcal{G}_{s+2}^{M'}(\mathcal{Y})}\left|\int g(u + u')K_{d,h_n}(u')\,d(\widehat{\nu}_n - \nu)(u)\,du'\right|$$

$$\leq \mathbb{E}\sup_{g \in \mathcal{G}_{s+2}^{cM'}(\mathcal{Y})}\left|\int g(u)\,d(\widehat{\nu}_n - \nu)(u)\right|$$

$$\leq D\inf\left\{a \geq \frac{cM'}{\sqrt{n}} : a \geq \frac{D}{\sqrt{n}}\int_a^{cM'}\sqrt{\log N_{[]}(\varepsilon, \mathcal{G}_{s+2}^{cM'}(\mathcal{Y}), L^2(\nu))}\,d\varepsilon\right\}, \qquad (C.28)$$

for some positive constant $D > 0$, where $N_{[]}(\varepsilon, \mathcal{G}_{s+2}^{cM'}(\mathcal{Y}), L^2(\nu))$ is the $\varepsilon$-bracketing entropy of the class of functions $\mathcal{G}_{s+2}^{cM'}(\mathcal{Y})$ with respect to the $L^2(\nu)$ norm. The last line follows from standard empirical process theory as used in the proof of Theorem 2.2; see (C.19). Note that by [24, Corollary 2.7.2], we have:

$$\log N_{[]}(\varepsilon, \mathcal{G}_{s+2}^{cL}(\mathcal{Y}^o), L^2(\nu)) \leq \gamma_d\left(\frac{1}{\varepsilon}\right)^{d/(s+2)}$$

for some $\gamma_d > 0$ depending only on dimension and the diameter of $\mathcal{Y}$.

We now plug-in the above bound into (C.28). By using $D_d$ to denote constants that change with $d$ and choosing $a = D_d n^{-1/2}$ for $2(s+2) > d$, $a = D_d n^{-1/2} \log(1+n)$ for $2(s+2) = d$ and $a = D_d n^{-(s+2)/d}$ for $2(s+2) < d$ in (C.28), we have:

$$\mathbb{E} \sup_{g \in \mathcal{G}_{s+2}^{M'}(\mathcal{Y})} \left| \int g(u+u') K_{d,h_n}(u') \, d(\widehat{\nu}_n - \nu)(u) \, du' \right| \lesssim r_{d,s}^{(n,n)}. \tag{C.29}$$

We now move on to the bounding the second term on the right hand side of (C.27). For this part, our main technical tool will be the classical arguments for smoothed empirical processes developed in [11]. Towards this direction, set $\overline{g}(u) = g(-u)$ (different from $\overline{g}_n(\cdot)$ defined earlier) for $g(\cdot) \in \mathcal{G}_{s+2}^{M'}$ and note that by [11, Lemma 4], we have:

$$\left| \int g(u+u') K_{d,h_n}(u') f_\nu(u) \, du \, du' - \int g(u) f_\nu(u) \, du \right| = \left| \int K_d(u) \left[ (\overline{g} * f_\nu)(h_n u) - (\overline{g} * f_\nu)(0) \right] \, du \right|, \tag{C.30}$$

where $(\overline{g} * f_\nu)(\cdot)$ is the standard convolution between $\overline{g}(\cdot)$ and $f_\nu(\cdot)$, and with a notational abuse $0$ denotes the $d$-dimensional zero vector. The important observation now is to note that $(\overline{g} * f_\nu)(\cdot)$ belongs to a higher order Sobolev class compared to $\overline{g}(\cdot)$ and $f_\nu(\cdot)$. In particular, as $f_\nu(\cdot) \in C^s(\mathcal{Y}; M)$ and $\overline{g}(\cdot) \in \mathcal{G}_{s+2}^{M'}(\mathcal{Y})$, we have $(\overline{g} * f_\nu)(\cdot) \in \mathcal{G}_{2s+2}^{M''}(\mathcal{Y})$ where $M''$ depends on both $M'$ and $M$.

Next, write $D^t(\overline{g} * f_\nu)(\cdot)$ to be the $t$-th derivative of $(\overline{g} * f_\nu)(\cdot)$ and note that by a multivariate Taylor's approximation

$$\int K_d(u) \left[ (\overline{g} * f_\nu)(h_n u) - (\overline{g} * f_\nu)(0) \right] \, du$$

$$= \int K_d(u) \sum_{r=1}^{2s+1} h_n^r \sum_{(i_1,i_2,\dots,i_r) \in \{1,2,\dots,d\}^r} [D^r(\overline{g} * f_\nu)(0)]_{i_1,\dots,i_r} u_{i_1} \dots u_{i_r} \, du + O(h_n^{2s+2}).$$

Recall that $K_d(u) = K(u_1) K(u_2) \dots K(u_d)$. As $K(\cdot)$ is of order $2s+2$ (see Assumption (A2)), all the integrals on the right hand side of the above display vanish. We then appeal to (C.30) to get:

$$\left| \int g(u+u') K_{d,h_n}(u') f_\nu(u) \, du \, du' - \int g(u) f_\nu(u) \, du \right| \lesssim h_n^{2s+2} = \lesssim n^{-\frac{2s+2}{d+2s}} (\log n)^{2s+2}.$$

We now compare the right hand side of the above display with $r_{d,s}^{(n,n)}$.

**When $d < 2(s+2)$:** $d + 2s < 4(s+1)$, and therefore $\frac{2s+2}{d+2s} > \frac{1}{2}$. This implies $n^{-\frac{2s+2}{d+2s}} (\log n)^{2s+2} \lesssim n^{-\frac{1}{2}} = r_{d,s}^{(n,n)}$.

**When $d = 2(s+2)$:** In this case $n^{-\frac{2s+2}{d+2s}} (\log n)^{2s+2} \lesssim n^{-\frac{1}{2}} (\log n)^{2s+2} \lesssim r_{d,s}^{(n,n)}$.

**When $d > 2(s+2)$:** Note that

$$\frac{2s+2}{d+2s} > \frac{s+2}{d} \Leftrightarrow 2ds + 2d > sd + 2s^2 + 2d + 4s \Leftrightarrow d > 2(s+2).$$

Therefore, once again $n^{-\frac{2s+2}{d+2s}} (\log n)^{2s+2} \lesssim n^{-\frac{s+2}{d}} = r_{d,s}^{(n,n)}$.

Therefore, combining the above observations, we have:

$$\left| \int g(u+u') K_{d,h_n}(u') f_\nu(u) \, du \, du' - \int g(u) f_\nu(u) \, du \right| \lesssim r_{d,s}^{(n,n)}.$$

Combining the above display with (C.29) establishes (C.25).

**Part 2.** This proof uses ideas from [14, Theorem 7], [20] and [2, Lemmas 2 and 3]. First recall all the notation introduced in Definition 2.4. Next, we will prove the following sequence of displays:

$$\limsup_{m,n \to \infty} \max_{k \le s} \max_{|\mathbf{m}|=k} \|\partial^{\mathbf{m}} \mathbb{E} \widehat{f}_\mu\|_{L^\infty(\widetilde{\mathcal{X}})} \le (T-1)M, \tag{C.31}$$

$$\limsup_{m,n\to\infty} \|\mathbb{E}\widehat{f}_\mu - f_\mu\|_{L^\infty(\widetilde{\mathcal{X}})} = 0 \tag{C.32}$$

$$\limsup_{m,n\to\infty} \mathbb{P}\left(\|\widehat{f}_\mu - \mathbb{E}\widehat{f}_\mu\|_{C^s(\widetilde{\mathcal{X}})} \geq \varepsilon\right) = 0, \tag{C.33}$$

for any arbitrary $\varepsilon > 0$ and $L^\infty(\mathcal{X})$ denotes the uniform norm on $\mathcal{X}$.

Clearly, (C.31), (C.32), and (C.33) together yield part 1 of the theorem.

*Proof of* (C.31). Observe that

$$\mathbb{E}\widehat{f}_\mu(x) = \frac{1}{h_m^d}\mathbb{E}K_d\left(\frac{x - X_1}{h_m}\right) = \frac{1}{h_m^d}\int K_d\left(\frac{x - z}{h_m}\right) f_\mu(z)\, dz. \tag{C.34}$$

Since the maximums taken in (C.31) are over finite sets, it suffices to show that for any fixed $\mathfrak{m}$ with $|\mathfrak{m}| \leq s$, we have:

$$\sup_{x\in\widetilde{\mathcal{X}}} |\partial^{\mathfrak{m}}\mathbb{E}\widehat{f}_\mu(x)| = \sup_{x\in\mathcal{X}}\left|\frac{1}{h_n^d}\int K_d\left(\frac{z}{h_n}\right)\partial^{\mathfrak{m}} f(x + z)\, dz\right| \leq (T - 1)M. \tag{C.35}$$

Here the first equality in the above display follows from (C.34) and Fubini's Theorem. Here $\partial^{\mathfrak{m}} f_\mu(\cdot)$ is defined in the weak sense, i.e., it is defined naturally in the interior of the support of $f_\mu(\cdot)$, denoted by $\mathcal{X}$; it is set to be 0 outside $\mathcal{X}$ and defined arbitrarily on the boundary of $\mathcal{X}$. Note that the definition on the boundary doesn't matter as we are integrating with respect to the Lebesgue measure and the boundary of $\mathcal{X}$ has Lebesgue measure 0.

Next note that, by (C.35), we have:

$$\sup_{x\in\widetilde{\mathcal{X}}} |\partial^{\mathfrak{m}}\mathbb{E}\widehat{f}_\mu(x)| \leq \|f_\mu\|_{C^s(\mathcal{X})}h_m^{-d}\int |K_d(z/h_m)|\, dz \leq (T - 1)\|f_\mu\|_{C^s(\mathcal{X})}.$$

This establishes (C.31).

*Proof of* (C.32). First note that, as $\widetilde{\mathcal{X}}$ is a compact subset of $\mathcal{X}^o$, there exists $\delta > 0$ such that

$$\widetilde{\mathcal{X}}_{\delta'} := \{x + z : \|z\| \leq \delta;\, , x \in \widetilde{\mathcal{X}}\} \subseteq \mathcal{X}^o \qquad \forall\, 0 < \delta' \leq \delta.$$

Clearly, $\widetilde{\mathcal{X}}_{\delta'}$ is compact for all $\delta' > 0$. Fix an arbitrary $\delta' \leq \delta$. By using (C.34) and a change of variable formula, we have:

$$\|\mathbb{E}\widehat{f}_\mu - f_\mu\|_{L^\infty(\widetilde{\mathcal{X}})} = \sup_{x\in\widetilde{\mathcal{X}}}\left|\frac{1}{h_m^d}\int K_d\left(\frac{z}{h_m}\right)(f(x + z) - f(x))\, dz\right|$$

$$\leq (T - 1)\sup_{x\in\widetilde{\mathcal{X}}}\sup_{\|z\|\leq\delta'} |f(x + z) - f(x)| + 2M\int_{\|z\|>\delta'h_m^{-1}} |K_d(z)|\, dz$$

$$\leq (T - 1)M\delta' + 2M\left(\frac{h_m}{\delta'}\right)^{2s+2}\int \|z\|^{2s+2}|K_d(z)|\, dz.$$

Observe that as $m, n \to \infty$, the second term on the right hand side of the above display converges to 0. This implies

$$\limsup_{m,n\to\infty}\|\mathbb{E}\widehat{f}_\mu - f_\mu\|_{L^\infty(\widetilde{\mathcal{X}})} \leq (T - 1)M\delta'.$$

As $\delta'$ can be chosen arbitrarily small, this completes the proof of (C.32).

*Proof of* (C.33). The main technical tool for this part is Lemma D.3 which we borrow from [2, Lemma 9] (also see [18, Theorem 4.1]). The proof is very similar to [2, Lemma 3]. Consider the following class of functions:

$$\mathcal{G} = \left\{g_x(z, h) :\ g_x(z, h) = \partial^{\mathfrak{m}}K_d\left(\frac{(x - z)}{h}\right),\ x \in \widetilde{\mathcal{X}},\ |\mathfrak{m}| \leq s\right\}.$$

Observe that

$$\sup_{|\mathfrak{m}|\leq s}\sup_{x\in\widetilde{\mathcal{X}}}\sup_{h\in(0,1)} h^{-d}\mathbb{E}\left[\partial^{\mathfrak{m}}K_d\left(\frac{(x - z)}{h}\right)\right]^2 \leq \|K\|_{C^s(\mathbb{R}^d)}\|f\|_{C^s(\mathcal{X})}\sup_{|\mathfrak{m}|\leq s}\int |\partial^{\mathfrak{m}}K_d(v)|\, dv < \infty.$$

Further, by Assumption (A2), $\partial^{\mathfrak{m}} K_d(\cdot)$ is differentiable for each $|\mathfrak{m}| \leq s$. Consequently $\mathcal{G}$ is point wise measurable and of VC-type (see [8, Lemma A.1]; also see [24, Section 2.6] for definitions of point wise differentiability and VC classes). This verifies the assumptions of Lemma D.3. Observe that

$$\frac{1}{n}\sum_{i=1}^{m}\partial^{\mathfrak{m}} K\left(\frac{x-X_i}{h_m}\right) = h_m^{d+|\mathfrak{m}|}\partial^{\mathfrak{m}}\widehat{f}_\mu(x), \qquad \mathbb{E}\left[\partial^{\mathfrak{m}} K\left(\frac{x-X}{h_m}\right)\right] = h_m^{d+|\mathfrak{m}|}\mathbb{E}\left[\partial^{\mathfrak{m}}\widehat{f}_\mu(x)\right].$$

A direct application of Lemma D.3 for all $|\mathfrak{m}| \leq s$, then implies

$$\sup_{x\in\widetilde{\mathcal{X}}}\sqrt{\frac{m}{h_m^d \log m}} \cdot h_m^{s+d}\|\widehat{f}_\mu - \mathbb{E}\widehat{f}_\mu\|_{C^s(\widetilde{\mathcal{X}})} = O_p(1).$$

Using the observation that $mh_m^{d+2s}/\log m \to 0$ as $m \to \infty$ then completes the proof. $\qquad \square$

*Proof of Proposition 2.6.* As $\mu \neq \nu$, we have $W_2(\mu,\nu) > 0$. Therefore,

$$|W_2(\widetilde{\mu}_m,\widetilde{\nu}_n) - W_2(\mu,\nu)| = \frac{W_2^2(\widetilde{\mu}_m,\widetilde{\nu}_n) - W_2^2(\mu,\nu)|}{W_2(\widetilde{\mu}_m,\widetilde{\nu}_n) + W_2(\mu,\nu)} \leq \frac{W_2^2(\widetilde{\mu}_m,\widetilde{\nu}_n) - W_2^2(\mu,\nu)|}{W_2(\mu,\nu)}.$$

The conclusion then follows from Theorem 2.5. $\qquad \square$

*Proof of Theorem 2.7.* Recall that $\widetilde{\mu}_m$ and $\widetilde{\nu}_n$ are defined as the empirical distributions induced by $M = n^{\frac{s+2}{2}}$ random samples drawn from $\widehat{f}_\mu$ and $\widehat{f}_\nu$ respectively, where $\widehat{f}_\mu, \widehat{f}_\nu$ are the kernel density estimates as presented in (2.5). Let us write $\mu_{h_n}$ and $\nu_{h_n}$ for the probability measure induced by the kernel density estimates $\widehat{f}_\mu$ and $\widehat{f}_\nu$ respectively. Once again, by using Theorem 2.2, (C.8), it suffices to prove the following:

$$\mathbb{E}\left|W_2^2(\widetilde{\mu}_m,\widetilde{\nu}_n) - W_2^2(\mu,\nu)\right|. \tag{C.36}$$

Next note that by the triangle inequality, (C.36) can be bounded above by:

$$\mathbb{E}\left|W_2^2(\widetilde{\mu}_m,\nu_{h_n}) - W_2^2(\widetilde{\mu}_m,\widetilde{\nu}_n)\right| + \mathbb{E}\left|W_2^2(\widetilde{\mu}_m,\nu_{h_n}) - W_2^2(\mu_{h_n},\nu_{h_n})\right|$$
$$+ \mathbb{E}\left|W_2^2(\mu_{h_n},\nu_{h_n}) - W_2^2(\widetilde{\mu}_m,\widetilde{\nu}_n)\right|. \tag{C.37}$$

Next note that, by Theorem 2.5, we have:

$$\mathbb{E}\left|W_2^2(\mu_{h_n},\nu_{h_n}) - W_2^2(\widetilde{\mu}_m,\widetilde{\nu}_n)\right| \lesssim r_{d,s}^{(n,n)}. \tag{C.38}$$

Next we show that

$$\mathbb{E}\left|W_2^2(\widetilde{\mu}_m,\nu_{h_n}) - W_2^2(\mu_{h_n},\nu_{h_n})\right| \lesssim r_{d,s}^{(n,n)}. \tag{C.39}$$

The other term in (C.37) can be bounded similarly.

Note that, conditioned on $X_1,\ldots,X_n,Y_1,\ldots,Y_n$, $\mu_{h_n}$ and $\nu_{h_n}$ are non-random measures and $\widetilde{\mu}_m$ and $\widetilde{\nu}_n$ are the empirical distributions on $M = n^{\frac{s+2}{2}}$ random samples from the measures $\mu_{h_n}$ and $\nu_{h_n}$, respectively. Therefore, conditioned on $X_1,\ldots,X_n,Y_1,\ldots,Y_n$ (which have fixed compact supports), we can invoke Corollary 2.3 to get:

$$\mathbb{E}\left|W_2^2(\widetilde{\mu}_m,\nu_{h_n}) - W_2^2(\mu_{h_n},\nu_{h_n})\right| \lesssim r_d^{(M,M)},$$

with $M = n^{\frac{s+2}{2}}$. Recall that:

$$r_d^{(M,M)} = \begin{cases} n^{-\frac{s+2}{4}} & \text{if } d \leq 3 \\ n^{-\frac{s+2}{4}}\log(1+n) & \text{if } d = 4 \\ n^{-\frac{s+2}{d}} & \text{if } d > 4 \end{cases}.$$

It therefore only remains to compare $r_d^{(M,M)}$ and $r_{d,s}^{(n,n)}$.

**Case 1:** $d \leq 2(s+2)$. In this case, if $d = 1,2,3$, then $r_d^{(M,M)} = n^{-\frac{s+2}{4}} = n^{-\frac{1}{2}} \times n^{-\frac{s}{4}} \lesssim n^{-\frac{1}{2}}$. If $d = 4$, then $r_d^{(M,M)} = n^{-\frac{s+2}{4}}\log(1+n) = n^{-\frac{1}{2}} \times \left(n^{-\frac{s}{4}}\log n\right) \lesssim n^{-\frac{1}{2}}$. If $d > 4$, then $r_d^{(M,M)} = n^{-\frac{s+2}{d}} \lesssim n^{-\frac{1}{2}}$ as $\frac{s+2}{d} \geq \frac{1}{2}$. Therefore, in all the cses, $r_d^{(M,M)} \lesssim n^{-\frac{1}{2}} = r_{d,s}^{(n,n)}$ for $d \leq 2(s+2)$.

**Case 2:** $d > 2(s+2)$. As $s > 0$, then $d > 4$. In this case, once again $r_d^{(M,M)} = n^{-\frac{s+2}{d}} = r_{d,s}^{(n,n)}$.

This establishes (C.39) and completes the proof. $\qquad \square$

## C.2 Proofs from Appendix B

*Proof of Theorem B.1.* First define the following measure:

$$\rho_0^{\mathrm{OR}} := \left(\frac{1}{2}\mathrm{Id} + \frac{1}{2}T_0\right) \# \widetilde{\mu}_m.$$

Fix any $\gamma \in \widetilde{\Gamma}_{\min}$. By applying the triangle inequality followed by a power mean inequality, we have:

$$\sup_{\gamma \in \widetilde{\Gamma}_{\min}} W_2^2\left(\widehat{\rho}_0^{\gamma}, \rho_0\right) \lesssim W_2^2\left(\rho_0^{\mathrm{OR}}, \rho_0\right) + \sup_{\gamma \in \widetilde{\Gamma}_{\min}} W_2^2\left(\widehat{\rho}_0^{\gamma}, \rho_0^{\mathrm{OR}}\right). \tag{C.40}$$

Next observe that $\rho_0^{\mathrm{OR}}$ is the empirical distribution corresponding to $m$ random samples drawn according to $\rho_0$. Therefore, by using [10, Theorem 1], we get:

$$W_2^2\left(\rho_0^{\mathrm{OR}}, \rho_0\right) \lesssim r_d^{(m,m)}. \tag{C.41}$$

Next we will bound the second term on the right hand side of (C.40). Towards this direction, recall the definition of $\Pi(\cdot, \cdot)$ from Section 1.1. Consider the following coupling:

$$\pi_0^{\gamma} := \left(\frac{1}{2}\mathrm{Id} + \frac{1}{2}\widetilde{T}_{m,n}^{\gamma}, \frac{1}{2}\mathrm{Id} + \frac{1}{2}T_0\right) \# \widetilde{\mu}_m.$$

Observe that $\pi_0^{\gamma} \in \Pi\left(\widehat{\rho}_0^{\gamma}, \rho_0^{\mathrm{OR}}\right)$. By plugging the coupling $\pi_0^{\gamma}$ into the definition of 2-Wasserstein distance in (1.3), we further get:

$$\sup_{\gamma \in \widetilde{\Gamma}_{\min}} W_2^2\left(\widehat{\rho}_0^{\gamma}, \rho_0^{\mathrm{OR}}\right) \leq \sup_{\gamma \in \widetilde{\Gamma}_{\min}} \int \|x - y\|^2 \, d\pi_0^{\gamma}(x, y)$$

$$= \sup_{\gamma \in \widetilde{\Gamma}_{\min}} \int \|\widetilde{T}_{m,n}^{\gamma}(x) - T_0(x)\|^2 \, d\widetilde{\mu}_m(x)$$

$$= O_p\left(r_d^{(m,n)} \times (\log\left(1 + \max\{m, n\}\right))^{t_{d,\alpha}}\right) \tag{C.42}$$

where the last inequality follows from Theorem 2.2. Combining (C.41) and (C.42) with (C.40) completes the proof. $\square$

*Proof of Theorem B.3.* Let $T_1^{(n)}(\cdot)$ and $T_2^{(n)}(\cdot)$ be the optimal transport maps from $\mu^{(n)}$ to $\upsilon_1$ and $\nu^{(n)}$ to $\upsilon_2$. Set

$$\widehat{x}_{ij}^{\mathrm{OR}} := K_1(T_1^{(n)}(X_i), T_1^{(n)}(X_j)), \qquad \widehat{y}_{ij}^{\mathrm{OR}} := K_2(T_2^{(n)}(Y_i), T_2^{(n)}(Y_j))$$

and define the oracle version of $\widehat{\mathrm{rHSIC}}$ as follows:

$$\widehat{\mathrm{rHSIC}}^{\mathrm{OR}} := \underbrace{n^{-2}\sum_{i,j} \widehat{x}_{ij}^{\mathrm{OR}}\widehat{y}_{ij}^{\mathrm{OR}}}_{\widehat{A}_{n,1}^{\mathrm{OR}}} + \underbrace{n^{-4}\sum_{i,j,r,s} \widehat{x}_{ij}^{\mathrm{OR}}\widehat{y}_{rs}^{\mathrm{OR}}}_{\widehat{A}_{n,2}^{\mathrm{OR}}} - \underbrace{2\,n^{-3}\sum_{i,j,r} \widehat{x}_{ij}^{\mathrm{OR}}\widehat{y}_{ir}^{\mathrm{OR}}}_{\widehat{A}_{n,3}^{\mathrm{OR}}}. \tag{C.43}$$

The proof of Theorem B.3 now proceeds using the following steps:

**Step I:** We show that:

$$\mathbb{E}\left|\widehat{\mathrm{rHSIC}}^{\mathrm{OR}} - \mathrm{rHSIC}(\pi^{(n)}|\mu^{(n)} \times \nu^{(n)})\right| \lesssim n^{-1/2}, \tag{C.44}$$

where $\widehat{\mathrm{rHSIC}}^{\mathrm{OR}}(\cdot|\cdot)$ is defined in (B.6).

**Step II:** We prove that:

$$\mathbb{E}\left|\widehat{\mathrm{rHSIC}}^{\mathrm{OR}} - \widehat{\mathrm{rHSIC}}\right| \lesssim \sqrt{r_d^{(n,n)}}. \tag{C.45}$$

**Step III:** We combine steps I and II to prove Theorem B.3. Let us begin with this step first. Note that by using the triangle inequality, we have:

$$\widehat{\mathrm{rHSIC}} \geq \mathrm{rHSIC}(\pi^{(n)}|\mu^{(n)} \times \nu^{(n)}) - \left|\widehat{\mathrm{rHSIC}}^{\mathrm{OR}} - \mathrm{rHSIC}\right| - \left|\widehat{\mathrm{rHSIC}}^{\mathrm{OR}} - \widehat{\mathrm{rHSIC}}\right|. \tag{C.46}$$

Next observe that by steps I and II,

$$\max\left\{\left|\widehat{\mathrm{rHSIC}}^{\mathrm{OR}} - \mathrm{rHSIC}\right|, \left|\widehat{\mathrm{rHSIC}}^{\mathrm{OR}} - \widehat{\mathrm{rHSIC}}\right|\right\} = O_p\left(\sqrt{r_{d_1,d_2}^{(n,n)}}\right).$$

Using the above display with (C.46) and the assumption $(r_{d_1,d_2}^{(n,n)})^{-1/2}\mathrm{rHSIC}(\pi^{(n)}|\mu^{(n)}\times\nu^{(n)}) \to \infty$, we have:

$$\left(r_{d_1,d_2}^{(n,n)}\right)^{-1/2}\widehat{\mathrm{rHSIC}} \xrightarrow{P} \infty.$$

Therefore, as $n\sqrt{r_{d_1,d_2}^{(n,n)}} \to \infty$ and $c_{n,\alpha} = O(1)$ (see [6, Theorem 4.1]), we have:

$$\mathbb{E}\phi_{n,\alpha} = \mathbb{P}(n \times \widehat{\mathrm{rHSIC}} \geq c_{n,\alpha}) \to 1$$

under $(r_{d_1,d_2}^{(n,n)})^{-1/2}\mathrm{rHSIC}(\pi^{(n)}|\mu^{(n)} \times \nu^{(n)}) \to \infty$. This completes the proof.

It therefore remains to prove steps I and II. For step I, let $(X_1', Y_1'), \ldots, (X_n', Y_n') \overset{i.i.d.}{\sim} \pi^{(n)}$. Fix an arbitrary $1 \leq j \leq n$. Let $\widehat{A}_{n,1,j}^{\mathrm{OR},'}$ be the same as $\widehat{A}_{n,1}^{\mathrm{OR}}$ except with $(X_j, Y_j)$ replaced by $(X_j', Y_j')$. It is easy to check by the compactness of supports of all distributions involved, that:

$$\max_{1 \leq j \leq n} |\widehat{A}_{n,1}^{\mathrm{OR}} - \widehat{A}_{n,1,j}^{\mathrm{OR},'}| \lesssim n^{-1}.$$

Therefore by using Mcdiarmid's inequality (see [3, Theorem 6.5]), we have, for any $t > 0$,

$$\mathbb{P}\left(\sqrt{n}(\widehat{A}_{n,1}^{\mathrm{OR}} - \mathbb{E}\widehat{A}_{n,1}^{\mathrm{OR}}) \geq t\right) \leq \exp(-Ct^2)$$

for some constant $C > 0$ free of $n$ and $t$. Similar concentrations can be derived for $\widehat{A}_{n,2}^{\mathrm{OR}}$ and $\widehat{A}_{n,3}^{\mathrm{OR}}$. Combining these concentrations with the observation that

$$\mathrm{rHSIC}(\pi^{(n)}|\mu^{(n)} \times \nu^{(n)}) = \mathbb{E}\widehat{A}_{n,1}^{\mathrm{OR}} + \mathbb{E}\widehat{A}_{n,2}^{\mathrm{OR}} - 2\mathbb{E}\widehat{A}_{n,3}^{\mathrm{OR}}$$

completes the proof of step I.

We now move on to step II. Recall the definition of $\widehat{\mathrm{rHSIC}}$ from (B.5) and write:

$$\widehat{\mathrm{rHSIC}} = \underbrace{n^{-2}\sum_{i,j}\widehat{x}_{ij}\widehat{y}_{ij}}_{\widehat{A}_{n,1}} + \underbrace{n^{-4}\sum_{i,j,r,s}\widehat{x}_{ij}\widehat{y}_{rs}}_{\widehat{A}_{n,2}} - \underbrace{2\,n^{-3}\sum_{i,j,r}\widehat{x}_{ij}\widehat{y}_{ir}}_{\widehat{A}_{n,3}}.$$

By the Lipschitzness of $K_1(\cdot,\cdot)$ and $K_2(\cdot,\cdot)$, we have:

$$\left|\widehat{x}_{ij} - \widehat{x}_{ij}^{\mathrm{OR}}\right| \lesssim \|\widehat{T}_{1,n}(X_i) - T_1^{(n)}(X_i)\| + \|\widehat{T}_{1,n}(X_j) - T_1^{(n)}(X_j)\|,$$

$$\left|\widehat{y}_{ij} - \widehat{y}_{ij}^{\mathrm{OR}}\right| \lesssim \|\widehat{T}_{2,n}(Y_i) - T_2^{(n)}(Y_i)\| + \|\widehat{T}_{2,n}(Y_j) - T_2^{(n)}(Y_j)\|.$$

Therefore, by using the fact that the probability measures $\upsilon_1$ and $\upsilon_2$ are compactly supported, we get:

$$\left|\widehat{A}_{n,1} - \widehat{A}_{n,1}^{\mathrm{OR}}\right| \lesssim \frac{1}{n}\sum_{i=1}^{n}\|\widehat{T}_{1,n}(X_i) - T_1^{(n)}(X_i)\| + \frac{1}{n}\sum_{j=1}^{n}\|\widehat{T}_{2,n}(Y_j) - T_2^{(n)}(Y_j)\|.$$

The same bound can similarly be verified for $|\widehat{A}_{n,2} - \widehat{A}_{n,2}^{\mathrm{OR}}|$ and $|\widehat{A}_{n,3} - \widehat{A}_{n,3}^{\mathrm{OR}}|$. Combining these observations, we have:

$$\left|\widehat{\mathrm{rHSIC}} - \widehat{\mathrm{rHSIC}}^{\mathrm{OR}}\right| \lesssim \frac{1}{n}\sum_{i=1}^{n}\|\widehat{T}_{1,n}(X_i) - T_1^{(n)}(X_i)\| + \frac{1}{n}\sum_{j=1}^{n}\|\widehat{T}_{2,n}(Y_j) - T_2^{(n)}(Y_j)\|$$

$$\leq \sqrt{\frac{1}{n}\sum_{i=1}^{n}\|\widehat{T}_{1,n}(X_i) - T_1^{(n)}(X_i)\|^2} + \sqrt{\frac{1}{n}\sum_{j=1}^{n}\|\widehat{T}_{2,n}(Y_j) - T_2^{(n)}(Y_j)\|^2}.$$

Step II then follows by invoking Corollary 2.3. $\qquad\square$

# D  Auxiliary definitions and results

**Definition D.1** (Subdifferential set and subgradient). *Given a convex function $f : \mathbb{R}^d \to \mathbb{R} \cup \{\infty\}$, we define the* subdifferential *set of $f(\cdot)$ at $x \in dom(f) := \{z \in \mathbb{R}^d : f(z) < \infty\}$ as follows:*

$$\partial f(x) := \{\xi \in \mathbb{R}^d :\ f(x) + \langle \xi, y - x \rangle \leq f(y), \quad \text{for all } y \in \mathbb{R}^d\}.$$

*Any element in the set $\partial f(x)$ is called a* subgradient *of $f(\cdot)$ at $x$.*

**Definition D.2** (Strong convexity). *A function $f : \mathbb{R}^d \to \mathbb{R} \cup \{\infty\}$ is strongly convex with parameter $\lambda > 0$, if, for all $x, y \in dom(f) = \{z \in \mathbb{R}^d : f(z) < \infty\}$, the following holds:*

$$f(y) \geq f(x) + \langle \xi_x, y - x \rangle + \frac{\lambda}{2} \|y - x\|^2,$$

*where $\xi_x \in \partial f(x)$, the subgradient of $f(\cdot)$ at $x$ as in Definition D.1.*

**Definition D.3** (Wavelet basis). *We present our main assumptions on the wavelet basis discussed in Appendix A only for the wavelets on the space $\mathcal{X}$. The same assumptions are also required for the wavelets on $\mathcal{Y}$. These are essentially a subset of the assumptions laid out in [131, Appendix E] as we heavily rely on [131, Theorem 1] for proving Theorem A.1.*

1. *(**Regularity**). Fix $r > \max\{s, 1\}$. The functions in $\boldsymbol{\Phi}$ and $\boldsymbol{\Psi}_j$, $j \geq 0$ have $r$ continuous derivatives, and all polynomials of degree at most $r$ on $\mathcal{X}$ lie in the span of the functions in $\boldsymbol{\Phi}$.*

2. *(**Tensor construction**). Each $\psi(\cdot) \in \boldsymbol{\Psi}_j$ can be expressed as $\psi(x) = \prod_{i=1}^d \psi_i(x_i)$, where $x = (x_1, \ldots, x_d)$, for some univariate functions $\psi_i(\cdot)$'s.*

3. *(**Locality**). For each $\psi(\cdot) \in \boldsymbol{\Psi}_j$ there exists a rectangle $I_\psi \subseteq \mathcal{X}$ such that $supp(\psi) \subseteq I_\psi$, $diam(I_\psi) \leq C_1 \cdot 2^{-j}$, and $\sup_{x \in \mathcal{X}} \sum_{\psi(\cdot) \in \boldsymbol{\Psi}_j} \mathbb{1}(x \in I_\psi) \leq C_2$ for some constants $C_1, C_2 > 0$.*

4. *(**Bernstein estimate**). $\|\nabla f\|_{L^2(\mathcal{X})} \leq C_3 \cdot 2^j \|f\|_{L^2(\mathcal{X})}$ for any $f(\cdot)$ in the span of the functions in $\mathrm{span}\,(\boldsymbol{\Phi} \cup \{\cup_{0 \leq k < j} \boldsymbol{\Psi}_j\})$. Here $C_3$ is some positive constant.*

**Lemma D.1** (Strong convexity and Lipschitzness, see [15]). *$\varphi_0^*(\cdot)$ is strongly convex with parameter $(1/L)$ if and only if $T_0(\cdot)$ is $L$-Lipschitz continuous.*

**Lemma D.2** (Gradient of dual). *Recall the definition of $f^*(\cdot)$ from (1.5) and $\partial f(\cdot)$ from Definition D.1. Then the following equivalence holds:*

$$\langle x, y \rangle = f(x) + f^*(y) \quad \Longleftrightarrow \quad y \in \partial f(x) \quad \Longleftrightarrow \quad x \in \partial f^*(y).$$

**Lemma D.3** (Bounding expected supremum of empirical process, see [2, 18]). *Let $f(\cdot)$ be a probability density supported on some subset of $\mathbb{R}^d$, and say $Z \sim f(\cdot)$. Let $\mathcal{G}$ be a class of uniformly bounded measurable functions from $\mathbb{R}^d \times (0, 1]$ to $\mathbb{R}$, such that:*

$$\sup_{g(\cdot) \in \mathcal{G}} \sup_{h \in (0,1]} h^{-d} \mathbb{E}[g^2(Z, h)] < \infty,$$

*and such that the class*

$$\mathcal{G}_0 := \{x \mapsto g(x, h) :\ g(\cdot) \in \mathcal{G},\ h \in (0, 1)\}$$

*is point wise measurable and of VC-type (see [24, Section 2.6] for relevant definitions of VC classes of sets/functions and point wise measurability). Then there exists $b_0 \in (0, 1)$ such that if $Z_1, Z_2, \ldots$ is an i.i.d. sequence of observations from the probability density $f(\cdot)$, we have:*

$$\sup_{g(\cdot) \in \mathcal{G}} \sup_{\frac{\log n}{n} \leq h^d \leq b_0} \sqrt{\frac{n}{h^d \log n}} \left| \frac{1}{n} \sum_{i=1}^n g(Z_i, h) - \mathbb{E}[g(Z, h)] \right| = O_p(1).$$