# OpenReview forum: "Rates of Estimation of Optimal Transport Maps using Plug-in Estimators via  Barycentric Projections"
_NeurIPS.cc/2021/Conference — NeurIPS 2021 Poster_

### Official Review · Reviewer_XWVX · 2021-07-06

**Rating:** 5
**Confidence:** 3

**Summary:**

This article proposes a study on the rates of estimation of Brenier maps using the plug in "barycentric mapping" estimator. As a consequence of this study authors obtain rates of convergence for $W_2^{2}$ in various context (depending on the regularity of the measures and on the setting: discrete-discrete of semi-discrete). They apply their results for Wasserstein barycenter estimation and Nonparametric independence testing.


**Limitations And Societal Impact:**

Authors did not provide any potential negative societal impact of their work however, it is difficult for me to foresee these impacts..

**Main Review:**

On the one hand, from a theoretical point of view, I find the different results very interesting  on the other hand I find that this article does not really fit in the scope of a machine learning conference. Moreover I think that the article could benefit from an improvement in its writing (see below).

Overall the introduction and the problem are well presented. The main result is also quite interesting: it justifies the use of barycentric mapping as an "approximation" of the optimal Monge map. This result has many practical consequences, especially when barycentric mapping is used as a proxy for the Monge map (as in [1,2,3] or in domain adaptation where barycentric mapping is at the heart of many methods e.g. [4]). This opens, in my opinion, many practical perspectives. It also offers new justifications for the recent results concerning the estimation of $W_2^{2}$.

My main criticism concerning this article is that it does not really fit, in my opinion, in the scope of a machine learning conference but is more adapted to a "mathematical journal" format such as Annals of Statistics. First, no experience is given in the paper. I understand the purpose of purely theoretical papers but I think that a ML paper should contain at least one or two small experiments validating/illustrating the different approaches when possible (see my last comment).

More importantly, the majority of the paper's contributions are in fact in supplementary material which is exclusively made up of proofs, quite specific and complex. So the paper is not really 8 pages long, but more like 20 pages of quite extensive mathematics. Consequently, given the time constraints inherent to NeurIPS, it seems to me quite reasonable to say that most ML reviewers cannot certify that all the proofs in such a paper are correct. Since these proofs are the main core of the paper, it is reasonable to think that it is difficult to judge the quality of the paper in such a setting. This argument would be invalid, however, if the above mentioned proofs were written with care and with a bit of hand-holding. However, the paper contains a lot of proofs without context or discussion and some "proofs by omission" based on other results.

Indeed, the proofs are not self-contained, making the task of reviewing them very difficult, even for someone used to handling the tools of optimal transport. There are many calls in the proofs to results/equations of other papers without discussion or context. In the different proofs of the supplementary, there are no less than 12 calls to other results, which is, I think, a lot ("see Proposition 2 in [16]", "see Theorem 2.10 in [7]", "Note that by [15, Equation 26]"). Although it is common to write a proof by appealing to other results, this is usually the case when the result is "somewhat known" or by explaining a bit the outline of these results. In many cases here, these results are very specific and are not discussed. It is also difficult to say, without reading the whole articles in question, if the assumptions of the different theorems referred to are really applicable in these specific cases. For example  [13, Theorem 17] for Step I of the supplementary (row 95) calls for a theorem which itself has many assumptions and it is difficult to see if these assumptions are validated here, because no explanation or context is given. Some other proofs are not very elaborated (row 79-80) "The general strategy to bound the term in (A.20) is derived from some intermediate steps in the proofs of [5, Lemmas 3 and 4]. We still present a sketch here for completeness."

Finally, the paper ends  on a theorem without any conclusion, which I find rather clumsy. No discussion or perspective is given about the "Application" part (which is actually another theoretical part). For example, it is rather difficult to understand the interest of the "Nonparametric independence testing" section: what is "the permutation principle as is necessary for the usual HSIC" ? Why understanding "the local power of $\phi_{n,\alpha}$; under "changing sequence of alternatives converging to the null" " is a practical, interesting problem ?
In this section, for example, it would have been valuable to include a small experiment showing the interest of the main result for this problem. Another question that could be interesting is the practical comparison between the barycentric mapping estimator compared to the "Sinkhorn divergence" one of (Chizat, 2020). It seems to me that it is somehow harder to compute the barycentric mapping, since this is a standard OT problem, while the Sinkhorn divergence is easier due to the regularization and the complexity of Sinkhorn’s algorithm. On the other hand, for real data, maybe the estimator based on the barycentric mapping would better behave ?

For all these reasons I rather recommend rejecting this paper. I find the results interesting but I don't think it is, as it stands, a suitable paper for a Machine Learning conference. I admit that some of my comments are quite subjective ("what a ML paper should be") so I am wiling to change my opinion.

[1] Michaël Perrot, Nicolas Courty, Rémi Flamary, Amaury Habrard. Mapping estimation for discrete optimal transport
[2] Vivien Seguy, Bharath Bhushan Damodaran, Rémi Flamary, Nicolas Courty, Antoine Rolet, Mathieu Blondel. Large-Scale Optimal Transport and Mapping Estimation
[3] Elsa Cazelles, Felipe Tobar, Joaquin Fontbona. Streaming computation of optimal weak transport barycenters
[4] Nicolas Courty, Rémi Flamary, Amaury Habrard, Alain Rakotomamonjy. Joint Distribution Optimal Transportation for Domain Adaptation

----- After Rebuttal -----

I thank the authors for their response. After reading the rebuttal I decided to increase my score to 5. I agree that this paper proposes interesting new theoretical results and I agree with the other reviewers on this point. However, I think it would benefit from a rewrite of the proofs (to be as self-content as possible since it is the core of the paper) and of the "Application" part which does not seem to me to be detailed enough as it is. I also think that this paper deserves a conclusion in order to give some perspective on the different results. That's why I don't increase my final grade more widely. However, I do not object to the acceptance of this paper.

**Time Spent Reviewing:**

5

---

> ### Author Response · Authors · 2021-08-10
> **Response to Reviewer XWVX**
>
> Thanks for your careful reading of the paper and the supplementary material. In the sequel, we use $\textbf{(C)}$  to denote your (paraphrased) comments and  $\textbf{(R)}$ to denote our response.
>
> $\textbf{(C).}$ My main criticism concerning this article is that it does not really fit, in my opinion, in the scope of a machine learning conference but is more adapted to a "mathematical journal" ...
>
> $\textbf{(R).}$ We certainly appreciate your concern. We would like to respond to this by saying that the closest paper to ours, and in fact the main motivation for our submission to Neurips, is the paper: L. Chizat, P. Roussillon, F. Léger, F-X. Vialard, G. Peyré. "Faster Wasserstein Distance Estimation with the Sinkhorn Divergence" which appeared in NeuRIPS in 2020. This is a theoretical paper in the same area (although with a numerical study section). Moreover https://nips.cc/Conferences/2021/CallForPapers specifies that NeurIPS welcomes papers on statistics. With the growing interest of theoretical OT in Statistics, we felt that our paper might be suitable for NeurIPS.
>
> $\textbf{(C).}$ Indeed, the proofs are not self-contained, making the task of reviewing them very difficult, even for someone used to handling the tools of optimal transport. There are many calls in the proofs to results/equations of other papers without discussion or context.
>
> $\textbf{(R).}$ We thank you for your careful reading of the paper including the supplementary file. We completely acknowledge that too many references to results/equations in other papers can severely impede the readability. Regardless of whether our paper is selected for publication, we can assure you that we will reduce the number of references to results/equations from other papers, verify the assumptions in all cited results with a lot more clarity and remove "proofs by omission" by providing more detailed explanations. This includes in particular, statements like "see Proposition 2 in [16]", "see Theorem 2.10 in [7]", "Note that by [15, Equation 26]" (to name a few from your review), all of which we will replace with proof outlines at the very least.
>
> $\textbf{(C).}$ ... Finally, the paper ends on a theorem without any conclusion, which I find rather clumsy. No discussion or perspective is given about the "Application" part (which is actually another theoretical part). For example, it is rather difficult to understand the interest of the "Nonparametric independence testing" section: what is "the permutation principle as is necessary for the usual HSIC" ?
>
> $\textbf{(R).}$ In terms of the main paper, we have also decided to move section 3.2 (which you have mentioned is "clumsy", without "discussion" or "perspective") to the supplementary file (if this paper is accepted). We totally acknowledge that this section requires further discussion on its importance from a statistical+machine learning perspective. We believe moving it to the supplementary file will provide us enough space to incorporate better explanations and motivations for the contents of this section We believe this will open up space to discuss our technical contributions + proof techniques in the main paper itself, which  can (to some extent) address your concern that the contributions of our paper are primarily in the supplementary file. Having said that, this paper is certainly theoretical and therefore, the supplementary material will continue to be quite lengthy and technical, which we believe does not violate any NeurIPS guidelines.

---

### Official Review · Reviewer_7wET · 2021-07-09

**Rating:** 7
**Confidence:** 3

**Summary:**

This paper studies the problem of estimating an optimal transport map between two probability distributions on Euclidean space, given IID samples from each of the distributions. The main result (Theorem 2.1) gives a general upper bound, in terms of the dual representation of the 2-Wasserstein distance. This is used to give upper bounds, in probability, on the rate of decay of the estimation risk both for the plugging in empirical distributions in the absence of smoothness assumptions and for kernel estimates under smoothness assumptions. Finally, implications for two applications (Wasserstein barycenter estimation and nonparametric independence testing) are discussed.

**Limitations And Societal Impact:**

The paper gives a reasonable discussion of its limitations. I don't foresee any potential negative societal impact of this work.

**Main Review:**

Overall, I think the paper's contributions are clearly sufficiently novel and relevant for acceptance. However, I think the paper would be rather difficult to read for all but a very small subset of the NeurIPS community.

Main Comments:
1) I found the paper rather notationally dense, and, in my opinion, the paper would benefit from providing more intuition in several places, especially around the definition (Definition 1.2) of the barycentric projection, the loss function (Eq. (1.8)), and the main result (Theorem 2.1). It would be helpful, for example, if there was some prose describing why Eq. (1.8) is a good loss function to upper bound, and what the roles of the terms appearing in Inequality (2.1) are.

2) In Theorem 2.4, it seems that the exact rate at which the bandwidth h_n vanishes does not affect the ultimate convergence rate, as long as it is slower than (log(n)/n)^{-1/(2s+d)}. It seems counterintuitive to me that a very large bandwidth, e.g. h_n = 1/log(n), would not slow the convergence rate. Could the authors provide any intuition for this phenomenon?

Minor Comments:
1) Line 40: The sentence "T_{m,n} converges at a rate of m^{-2/d} + n^{-2/d}" isn't really meaningful unless a loss or error metric is specified. Especially for nonparametric problems such as this, different metrics likely lead to different convergence rates. Hence, I suggest adding a brief description of the error metric here, or at least pointing the reader to Eq. (1.8).
2) In several places, such as Eqs. (2.1) and (2.2), it is impossible to visually distinguish the \overline's and \tilde's over \mu and \nu without zooming in quite a bit. This left me confused for a while trying to figure out the difference. Is it possible to make these notations more visually distinct?
3) Line 78: Typo: "distributions on X_1, . . . , X_m" -> "distributions of X_1, . . . , X_m"
4) Could the authors comment on how specific the results in this paper are to 2-Wasserstein, as opposed to general p-Wasserstein, distances?

**Time Spent Reviewing:**

3

---

> ### Author Response · Authors · 2021-08-10
> **Response to Reviewer 7wET**
>
> Thanks for your careful reading of the paper. In the sequel, we use $\textbf{(C)}$  to denote your (paraphrased) comments and  $\textbf{(R)}$ to denote our response.
>
> $\textbf{(C).}$ I found the paper rather notationally dense, and, in my opinion, the paper would benefit from providing more intuition in several places ...
>
> $\textbf{(R).}$ Thank you for your careful reading and pointing out places that require further explanation. We will add these details in subsequent versions. Given the space constraints, we have decided that we will move Section 3.2 to the supplementary material (if the paper is finally accepted by NeuRIPS) and use the additional space to add computational details, motivations and further explanations as you suggested.
>
> $\textbf{(C).}$ In Theorem 2.4, it seems that the exact rate at which the bandwidth $h_n$ vanishes does not affect the ultimate convergence rate.
>
> $\textbf{(R).}$ You are absolutely right. This is an aspect which we got sloppy with and apologise for. The bandwidth should be chosen exactly of the order $n^{-\frac{1}{2s+d}}$ upto logarithmic factors (not just slower than that threshold). This is particularly necessary to handle the bias term in the rate computation, in particular, to obtain A.26 in the supplementary file. We have since observed this mistake and made the appropriate correction. We will add all the relevant steps in the subsequent versions of our paper.
>
> $\textbf{(C).}$ Minor comments 1,2,3.
>
> $\textbf{(R).}$ We will address these accordingly.
>
> $\textbf{(C).}$ Could the authors comment on how specific the results in this paper are to 2-Wasserstein, as opposed to general p-Wasserstein, distances?
>
> $\textbf{(R).}$ We believe that the rates obtained for the $W_2$ distance (as stated in equation 1,9) are indeed generalizable to general $W_p$ distances. However, getting similar analogues of equation 1.8 beyond the $W_2$ loss function are much harder and, we believe, best left for future research.

---

> > ### Comment · Reviewer_7wET · 2021-08-31
> > **Thanks for author response**
> >
> > Thanks to the authors for their response. I believe the authors would be able to address my main concerns (regarding clarity) in minor revisions. Overall, I still lean towards acceptance (and am keeping my current score), but am open to rejecting due to the difficulty of reading the proofs, as suggested by other reviewers.

---

### Official Review · Reviewer_NLrD · 2021-07-13

**Rating:** 7
**Confidence:** 4

**Summary:**

In standard OT problem, marginal distributions of the data are unknown. The current work considers plug-in estimation using Barycentric projection and derived rate of estimation. The idea of Barycentric projection is to plug the estimated marginal distributions of the data in the OT problem. Conditional mean of given us estimated and rate of estimation is derived. Various plug-in estimators of density of are considered such as empirical CDF and kernel density estimator.

**Ethics Review Area:**

["I don’t know"]

**Limitations And Societal Impact:**

The concern is why the smoothness of the densities of are the same? In practice, the densities may have different smoothness degrees. Meanwhile, how to make it adaptive since selection of is a problem.

Another concern is the motivation of using OT for estimating conditional mean, compared with various nonparametric approaches in literature such as local polynomial, smoothing spline, etc.

**Main Review:**

The work is new and results are strong and important. Paper written is clear and of good quality.

**Time Spent Reviewing:**

3 hours

---

> ### Author Response · Authors · 2021-08-10
> **Response to Reviewer NLrD**
>
> Thanks for your careful reading of the paper. In the sequel, we use $\textbf{(C)}$  to denote your (paraphrased) comments and  $\textbf{(R)}$ to denote our response.
>
> $\textbf{(C).}$ The concern is why the smoothness of the densities of are the same? In practice, the densities may have different smoothness degrees. Meanwhile, how to make it adaptive since selection of is a problem.
>
> $\textbf{(R).}$ You are very right. The two densities need not have the same smoothness and in general, the rates of convergence results hold with $s$ replaced by the minimum level of smoothness among the two densities. With regards to the choice of $s$, the standard practice is to use Lepski's method. The same approach was also recommended in the paper [69] as cited. We will discuss this briefly in future versions.
>
> $\textbf{(C).}$ Another concern is the motivation of using OT for estimating conditional mean, compared with various nonparametric approaches in literature such as local polynomial, smoothing spline, etc.
>
> $\textbf{(R).}$ This is a very interesting point. Our reasoning behind using the conditional mean is that it is completely tuning free (i.e., does not require the choice of smoothing parameters like smoothing bandwidths) and can directly obtained by solving the discrete-discrete OT linear program. Of course, when working with continuous densities, an OT map exists and there is no need to compute conditional means.

---

### Official Review · Reviewer_TNQz · 2021-07-16

**Rating:** 7
**Confidence:** 4

**Summary:**

This paper analyze the statistical behavior of the general plug-in estimators of OT defined by barycentric projections. The authors provide a thorough analysis of the rate of convergence for the transport cost and the transport map. They also consider kernel smoothed plug-in estimators and relate its rate of convergence to the smoothness of the densities, which alleviates the curse of dimensionality suffered by the plug-in estimator.

**Limitations And Societal Impact:**

See comments above for suggestions.

**Main Review:**

Strength:

1. The upper bound on the stability of the transport map defined via barycentric projections is new. The rate of convergence of the smoothed plug-in estimators is quite interesting since it reflects the smoothness of the densities.
2. The paper has an extensive related work section discussing clearly the relationship of their work to previous ones.
3. The paper is technically solid. The theoretical results are well explained and the intuition are nicely conveyed.
4. The authors also design a discretized version of the smoothed plug-in estimator which enjoys the same rate of convergence and can be computed in practice.

Major comments:

1. In the construction of discretized plug-in estimator on line 255, the procedure to obtain samples from the kernel density estimators is not discussed. This should also appear in the complexity analysis of computing $\tilde T_{n,m}$ especially when the ambient dimension is high.
2. In Theorem 2.6, the number of samples M is set to be $n^{\frac{s+2}{2}}$. In practice, the smoothness s is unknown. It would be better to discuss how to select s in practice.
3. An important selling point of this paper is, in my opinion, a discretized plug-in estimator which enjoys a good rate of convergence and can be computed in practice. It is expected to see some experimental results supporting this claim.


Minor comments:

1. On line 227 the squared 2-Wasserstein distance is considered, while the square is dropped in Proposition 2.5.
2. Section 3.2 feels rushed. It would be better to explain more on the intuition behind rHSIC.

-----
Updates: the authors addressed most of my concerns in the response. While I do believe adding the experiments on the discretized plug-in estimator will be a good addition to the paper, I also think the current version is already a good paper. Hence, I am keeping my score.

**Time Spent Reviewing:**

6

---

> ### Author Response · Authors · 2021-08-10
> **Response to reviewer TNQz**
>
> Thanks for your careful reading of the paper. In the sequel, we use $\textbf{(C)}$  to denote your (paraphrased) comments and  $\textbf{(R)}$ to denote our response.
>
> $\textbf{(C).}$ In the construction of discretized plug-in estimator on line 255, the procedure to obtain samples from the kernel density estimators is not discussed.
>
> $\textbf{(R).}$ We will add more details on this computational aspect in later versions. Given the space constraints, we decided to push Section 3.2 to the supplementary material and use the additional space to add computational details, motivations and further explanations. In the supplementary file we will also add more explanations in Section 3.2 so as to ensure it doesn't feel rushed.
>
> $\textbf{(C)}$ ... An important selling point of this paper is, in my opinion, a discretized plug-in estimator which enjoys a good rate of convergence and can be computed in practice. It is expected to see some experimental results supporting this claim.
>
> $\textbf{(R).}$ This is a very interesting point to explore. However given the space constraints, we believe it will be too difficult to accommodate numerical experiments in this paper. We are in the process of writing another paper regarding estimation of OT maps and we believe detailed numerical experiments are a better fit there.
>
> $\textbf{(C).}$ In Theorem 2.6, the number of samples M is set to be $n^{\frac{s+2}{d}}$. In practice, the smoothness s is unknown. It would be better to discuss how to select s in practice.
>
> $\textbf{(R).}$ That is a very interesting question. To the best of our understanding, the simplest approach would be to use Lepski's method (see the paper "An alternative point of view on Lepski's method" by L. Birge). The same approach was also recommended in the paper [69] as cited. We will discuss this briefly in future versions.
>
> $\textbf{(C).}$ Minor comments.
>
> $\textbf{(R).}$ We will make these corrections in future versions.

---

> > ### Comment · Reviewer_TNQz · 2021-08-25
> > **Thank you for your response**
> >
> > Thank you for your response. Most of my concerns was addressed in the response. However, I do believe adding the experiments on the discretized plug-in estimator will be a good addition to the paper. Hence, I am keeping my score.

---

### Official Review · Reviewer_yWvi · 2021-07-29

**Rating:** 6
**Confidence:** 4

**Summary:**

Summary
=======

Context:
--------

This paper tackles the problem of estimating the rate of convergence of empirical Monge maps
(through its Kantorovich barycentric map approximator) to the true Monge map for
absolutely continuous Lebesgue measures.


Contribution
------------

- The authors provide new rates of convergence of the barycentric map using standard
empirical plug-in estimators (O(n^{-2/d}))
- If the measures are regular enough (Holder smooth), the dependence on the dimension
completely vanishes (which was established for Gaussian distributions https://arxiv.org/pdf/1905.10155.pdf)
- The authors propose to take advantage of this smoothness result to sample from smoothened distributions
(using a Kernel density estimator) and provide the sufficient conditions to obtain a desired rate. Higher
smoothness however requires more samples: a tradeoff between statistical accuracy and computational cost
cannot be avoided.



**Main Review:**

review
======

General assessment
------------------

For a purely theoretical paper, this paper is vey well structured and easy to read. Transitions are smooth, and theorems
are fairly well motivated and interpreted. Even though one may wonder whether a journal would have been more appropriate
for this paper, the theoretical contributions of this paper are novel and are certainly valuable for the OT community at Neurips.
My only main concern is the lack of clarity of how can proposition 2.5 hold, which suggests that the rate of convergence of W2
is significantly different when mu = nu (see point (5) below). This is the only point -- pre-rebuttal -- retaining me from increasing my score.
I would appreciate if the authors could shed some light on this: are there any assumptions mismatch between this setting and the one from [119] that would explain this disparity ? Otherwise shifting the samples of a distribution slightly would lead to better convergence rates.

Specific comments
-----------------

1) In remark 2.1, the authors mention that theorem 2.1 doesn't require compactly supported measures
unlike previous results (of [69]). Still, for all the derived convergence rates (subsequent theorems), at least one
of the  measures needs to have a compact support for a uniform bound of the potential function. While requiring less
assumptions on the measures to obtain similar stability results is certainly an improvement over the literature, I'm wondering
whether Theorem 2.1 can be used to obtain convergence rates (or any other simpler and interpretable upper bounds)
without assuming compacity of any of the measures.

2) Still on compacity: how would an assumption on the tail of the measures compare with the compacity assumption ? Could
the uniform norm be substitued with a less stringent norm in the technical derivations so as to generalize the known convergence
rates for Gaussians for example ?

3) Throughout the paper, T0 is assumed L-lipschitz. However the regularity of T0 can also be seen as a consequence of
the regularity of the measures themselves which begs the question: are there any sufficient conditions on the measures
that guarantee an L-lipschitz T0 without assuming "too much" smoothness on the measures.

4) The constant C in the corollary seems to be obtained through existence theorems of arbitrary constants C1 and C2. Do we have
any bounds on these constants and their dependence on the parameters of the problem ? Perhaps the diameter of the bounded supports ?

5) in L 248, authors mention that T_m,n of theorem 2.4 cannot be computed from the data, is this because T_m,n is a continuous
barycentric map derived from the continuous measures given by the KDE ? If so, I think the dependence on m,n should be slightly different
(in terms of notation) to distinguish it from the empirical map.

6) I find the result of proposition 2.5 quite intriguing. [119] provided a lower bound on the convergence rate for the case mu = nu,
that is a rate that cannot be beaten by any estimator. Prop 2.5 shows that this rate can be improved (significantly) if mu != nu.
This suggests that If I take a measure mu and shift its mean by a certain
value, I would need less samples to approximate the W2 value. Am I missing something here ?

7) A minor remark on the appendix: please use a different notation for \bar{nu}, distinguishing it from  \tilde{nu}
is quite challenging specially when verifying the proofs.


**Time Spent Reviewing:**

8

---

> ### Author Response · Authors · 2021-08-10
> **Response to Reviewer yWvi**
>
> Thanks for your careful reading of the paper. In the sequel, we use $\textbf{(C)}$ to denote your (paraphrased) comments and $\textbf{(R)}$ to denote our response.
>
> $\textbf{(C)}$: My only main concern is the lack of clarity of how can proposition 2.5 hold, which suggests that the rate of convergence of W2 is significantly different when $\mu = \nu$ (same as point 6 in the review).
>
> $(\textbf{R})$: The fact that $W_2(\mu,\nu)$ can be estimated fast when $\mu\neq\nu$ has been observed in this NeurIPS paper [31] https://papers.nips.cc/paper/2020/file/17f98ddf040204eda0af36a108cbdea4-Paper.pdf (Corollary 1) when no smoothness is assumed on $\mu$ and $\nu$. Under no smoothness the difference $W_2(\hat{\mu}_n,\hat{\nu}_n)-W_2(\mu,\nu)$ ($\hat{\mu}_n$, $\hat{\nu}_n$ are standard empirical measures) has a $n^{-2/d}$ convergence rate for $\mu\neq\nu$ whereas it is well known that $W_2(\hat{\mu}_n,{\mu})$ has a convergence rate of $n^{-1/d}$ (which can't be improved). Proposition 2.5 in our paper can be viewed as an extension of the aforementioned result to the case of smooth densities (which also leads to faster rates of convergence).
>
> With regards to rates of convergence under location shifts, it is indeed correct that if $\mu$ is a shift of $\nu$, then by proposition 2.5 (under appropriate smoothness), $W_2(\tilde{\mu}_n,\tilde{\nu}_n)-W_2(\mu,\nu)$ ($\tilde{\mu}_n$, $\tilde{\nu}_n$ are the smoothed empirical measures) has a $n^{-\frac{s+2}{d}}$ rate of convergence ($s$ being the degree of smoothness). Note that,
> upper bounds on $W_2(\tilde{\mu}_n,\mu)$ and $W_2(\tilde{\nu},\nu)$ naturally lead to an upper bound on the difference $W_2(\tilde{\mu}_n,\tilde{\nu}_n)-W_2(\mu,\nu)$ (by an application of the triangle inequality), whereas the opposite is not true. In particular, if for some $\mu_n$ and $\nu_n$, we have $W_2(\mu_n,\nu_n)-W_2(\mu,\nu)\overset{P}{\longrightarrow} 0$, it does not imply that $W_2(\mu_n,\mu)\overset{P}{\longrightarrow} 0$ or $W_2(\nu_n,\nu)\overset{P}{\longrightarrow} 0$ (it is easy to construct such examples).
>
> The above discussion shows that the problem of obtaining rates for $W_2((\cdot),\mu)$ with an appropriate estimator $(\cdot)$ is a harder problem than estimating $W_2(\mu,\nu)$. Consequently the minimax achievable lower bounds proved in [117] (http://proceedings.mlr.press/v99/weed19a.html, which we presume is the paper you were referring to) are slower than the rates obtained in Proposition 2.5 (and [31]).
>
> We will certainly ensure that we make the discussion after Proposition 2.5 clearer in future versions of the paper, so as to avoid this confusion.
>
> $\textbf{(C).}$ ... for all the derived convergence rates (subsequent theorems), at least one of the measures needs to have a compact support. how would an assumption on the tail of the measures compare with the compacity assumption ? Could the uniform norm be substituted with a less stringent norm in the technical derivations so as to generalize the known convergence rates for Gaussians for example ?
>
> $\textbf{(R).}$ You are indeed correct. It is possible to remove compactness assumption on both the measures using tools from empirical process theory. They can be replaced by moment assumptions that control the tail of the distribution and rates of convergence can be obtained based on the number of moments assumed finite. The primary technique needed to deal with case where the support of $\nu$ too is unbounded (we already cover the case where support of $\mu$ is unbounded in the paper) is to break the support into "shells" and apply Dudley's chaining technique (which we use in the proofs of all our main results) on each shell. Under appropriate tail decay conditions, it can be shown that only a few shells contribute whereas the others are negligible. This technique is motivated from the proof of Corollary 2.7.4 in the book ``Weak Convergence and Empirical processes". In the Gaussian case, this technique could be used to obtain dimension-free rates by exploiting the fact that Gaussian densities are infinitely smooth and admit finite exponential moments.
>
> However, we believe that the main focus of the paper is to show the flexibility and usefulness of Theorem 2.1. We do this by using Theorem 2.1 to obtain rates of convergence in the discrete + semi-discrete settings, and also to get better rates in the presence of smoothness.  Therefore we decided not to provide more complicated presentations of Theorems 2.3 and 2.4 with tail assumptions to keep the focus of the paper more streamlined. Further, given the page restrictions, we believe that this question of compacity is best addressed in future research.
>
> $\textbf{(C).}$ ... are there any sufficient conditions on the measures that guarantee an $L$-lipschitz T0 without assuming "too much" smoothness on the measures.
>
> $\textbf{(R).}$ There are indeed sufficient conditions ensuring Lipschitzness of transport maps thanks to the extensive work by Luis Caffarelli [21,22,23] and Alessio Figalli. One of the sufficient conditions we like would be that in https://arxiv.org/pdf/1806.01238.pdf Proposition 2.3. It is one of the few conditions we know that does not assume $\mu$ to be compactly supported.
>
> $\textbf{(C).}$ ... Do we have any bounds on these constants and their dependence on the parameters of the problem ?
>
> $\textbf{(R).}$ That is an interesting question but we believe that tracking the constants explicitly in terms of dimension will be very hard here. Particularly because the metric entropy bounds used from [65] themselves involve constants that are not explicit in terms of dimension.
>
> $\textbf{(C).}$ ... Notational clarifications in points 5 and 7.
>
> $\textbf{(R).}$ You are indeed right that the $T_{m,n}$ from theorem 2.4 cannot be computed as it is the barycentric projection between continuous densities. We will incorporate the notational clarifications you pointed out  in future versions of our paper.

---

> > ### Comment · Reviewer_yWvi · 2021-09-01
> > **response**
> >
> > Thanks for the detailed rebuttal. All my main concerns were addressed here. This paper is a valuable contribution to the field nonetheless I invite the authors to go make their proofs more accessible and self contained. A detailed sketch of proof could also help the reader grasp the reasoning behind the demonstrations without going through the technical derivations.

---

### Decision · Program_Chairs · 2021-09-27

**Decision:**

Accept (Poster)

**Comment:**

All the reviewers insisted on the high quality of the theoretical results, even if some numerical simulations would have been welcome (these could have been provided in the supplementary material). They also appreciated the quality of the rebuttal, which clarified some important questions. Some raised the issue that the part of the proof in the supplementary was lacking clarity and details, and this should be improved in the final version. I thus strongly urge the authors to take into account the feedback of the reviewers to improve the quality of the paper. For these reasons (lacks of numerical simulation and clarity of the proofs), after a discussion with the reviewers and in agreement with them, I decided to support acceptance but not for a spotlight.